# Transcriptional downregulation of MHC class I and melanoma de- differentiation in resistance to PD-1 inhibition

Jenny H. Lee[1,2,3], Elena Shklovskaya[1,2], Su Yin Lim[1,2], Matteo S. Carlino[2,4], Alexander M. Menzies[2,5], Ashleigh Stewart[1,2], Bernadette Pedersen[1,2], Malama Irvine[1,2], Sara Alavi[2,6], Jean Y. H. Yang [7,8], Dario Strbenac[7,8], Robyn P. M. Saw [2,9,10], John F. Thompson [2,9,10], James S. Wilmott [2,8,10], Richard A. Scolyer[2,8,10,11], Georgina V. Long[2,5,8,10], Richard F. Kefford[1,2] & Helen Rizos [1,2✉]

Transcriptomic signatures designed to predict melanoma patient responses to PD-1 blockade have been reported but rarely validated. We now show that intra-patient heterogeneity of tumor responses to PD-1 inhibition limit the predictive performance of these signatures. We reasoned that resistance mechanisms will reflect the tumor microenvironment, and thus we examined PD-1 inhibitor resistance relative to T-cell activity in 94 melanoma tumors collected at baseline and at time of PD-1 inhibitor progression. Tumors were analyzed using RNA sequencing and flow cytometry, and validated functionally. These analyses confirm that major histocompatibility complex (MHC) class I downregulation is a hallmark of resistance to PD-1 inhibitors and is associated with the MITF$^{low}$/AXL$^{high}$ de-differentiated phenotype and cancer-associated fibroblast signatures. We demonstrate that TGFß drives the treatment resistant phenotype (MITF$^{low}$/AXL$^{high}$) and contributes to MHC class I downregulation in melanoma. Combinations of anti-PD-1 with drugs that target the TGFß signaling pathway and/or which reverse melanoma de-differentiation may be effective future therapeutic strategies.

[1] Macquarie University, Sydney, NSW, Australia. [2] Melanoma Institute Australia, Sydney, NSW, Australia. [3] Chris O'Brien Lifehouse, Camperdown, NSW, Australia. [4] Crown Princess Mary Cancer Centre, Westmead and Blacktown Hospitals, Sydney, NSW, Australia. [5] Northern Sydney Cancer Centre, Royal North Shore Hospital, Sydney, NSW, Australia. [6] Melanoma Oncology and Immunology, Centenary Institute, Camperdown, NSW, Australia. [7] School of Mathematics and Statistics, The University of Sydney, Sydney, NSW, Australia. [8] Charles Perkins Centre, The University of Sydney, Sydney, NSW, Australia. [9] Department of Melanoma and Surgical Oncology, Royal Prince Alfred Hospital, Camperdown, NSW, Australia. [10] Sydney Medical School, The University of Sydney, Sydney, NSW, Australia. [11] Department of Tissue Pathology and Diagnostic Oncology, Royal Prince Alfred Hospital, Camperdown, NSW, Australia. ✉email: helen.rizos@mq.edu.au

Immunotherapy has transformed the treatment of melanoma patients, with monoclonal antibodies blocking the immune regulator programmed cell death protein 1 (PD-1; pembrolizumab and nivolumab) receiving rapid FDA approval following demonstration of a survival benefit compared with chemotherapy and the cytotoxic T-lymphocyte-associated protein 4 (CTLA-4) inhibitor, ipilimumab[1–3]. Despite durability of response, innate resistance to PD-1 inhibition occurs in 30% of melanoma patients[4–6] and approximately 25% of responding patients will develop acquired resistance, defined as disease progression following initial objective response, within two years of PD-1 inhibitor treatment[4].

The mechanisms responsible for failure of PD-1 inhibition are diverse and incompletely understood, although resistance effectors have been identified in a small subset of patients. These include the expression of alternate immune checkpoint inhibitors (TIM-3 and LAG-3)[7,8], loss of major histocompatibility complex (MHC) class I expression, abnormalities in the interferon-γ (IFNγ) immune effector signaling pathway (JAK1, JAK2, IFNGR1, STAT1 mutations)[9–13], oncogenic signaling (elevated ß-catenin/WNT) that leads to immune exclusion[14], T-cell induced secretion of immunosuppressive colony-stimulating factor 1[15] and an hypoxic tumor micro-environment that may impair T-cell function[16]. Furthermore, several immune and gene-expression signatures predictive of PD-1 inhibitor response have been reported, but few have been validated in independent patient cohorts[11,17–19]. For example, the innate PD-1 inhibitor resistance (IPRES) signature, which includes 26 gene signatures associated with de-differentiation and BRAF/MEK inhibitor resistance, was associated with lack of PD-1 inhibitor response in pre-treatment melanoma biopsies in one study[17], but was not associated with PD-1 inhibitor response in other melanoma cohorts[11,19].

In this study, we perform transcriptome and flow cytometric analysis on 94 longitudinal melanoma biopsies in a large cohort of melanoma patients receiving PD-1 inhibitors. Analysis of pre-treatment and on-treatment tumors, including those responding to therapy (RES) and those that progressed (PROG) due to innate or acquired resistance. We provide insights into the complex and heterogeneous response of individual metastases to PD-1 inhibition and the heterogeneous immune transcriptome profile observed in synchronous and longitudinal biopsies. In addition, we demonstrate that down-regulation of MHC class I expression, rather than complete loss of MHC class I molecules, is common in melanoma and potently driven by TGFß signaling and de-differentiation.

## Results

**Patient and tumor characteristics.** Transcriptome analysis was performed on RNA sequence data ($n = 79$ tumors; 55 patients) and flow cytometric analysis on single cell suspensions ($n = 31$; 24 patients) from a total of 94 melanoma tumors derived from 68 patients treated with PD-1 inhibitor monotherapy (Supplementary Data 1). In total 53 tumor biopsies were pre-treatment (PRE) and 41 were taken while on-treatment (Fig. 1A). PRE tumors from patients who subsequently underwent complete response (CR) or partial response (PR) by irRC criteria[20] were termed responding-PREs ($n = 31$), and PRE biopsies from patients who had stable disease (SD) or progressive disease (PD) by irRC ($n = 22$) were termed non-responding-PREs. Of the 41 on-treatment tumor specimens, six biopsies were taken from clinically responding lesions (RES) and 35 were biopsies taken at time of progression (PROG). All PROG lesions were characterized as either innate PROG tumors ($n = 22$) or acquired PROG tumors ($n = 13$). Innate PROGs were defined as pre-existing metastases that did not undergo tumor shrinkage or new metastases

identified within 6 months of starting treatment, and acquired PROGs were defined as pre-existing tumors that initially responded but subsequently progressed on PD-1 inhibitor or new metastases identified after 6 months of starting treatment (Fig. 1A).

The median patient age was 67 years (range 38–88) and 23/68 (34%) patients had received prior MAPK inhibitor therapy (Table 1). Of the 68 patients, 41 (60%) had a pre-treatment biopsy only, 15 (22%) had an on-treatment biopsy only and 12 (18%) patients had matching pre- and on-treatment biopsies available for analysis (Fig. 1B).

**Heterogeneous tumor responses and predicting anti-PD-1 response.** We initially examined the predictive accuracy of seven transcriptome signatures associated with clinical response to PD-1 inhibition[11,15,17–19,21,22] in the 44 PRE tumors with available RNA sequence data (Supplementary Data 1). None of the published immune-predictive signatures, including signatures indicative of inflammation (i.e., CD8+ T-cell, CYT score and the 18-immune gene set) accurately defined responding (CR/PR) or non-responding patients (SD/PD) in our cohort (Fig. 1C, Supplementary Data 2). Additionally, we found that none of these predictive transcriptomic signatures were consistently and significantly associated with irRC response in three separate immune-checkpoint inhibitor treated melanoma patient RNA-seq datasets (Supplementary Figure 1). Further, we did not detect any significant differentially expressed genes (FDR-adjusted p-values <0.05) or gene signatures (ssGSEA score differences between two groups; FDR-adjusted p-values <0.05) between responders and non-responders in our cohort.

We hypothesized that accurately predicting melanoma patient response to PD-1 inhibitor therapy based solely on the characteristics of a single pre-treatment biopsy may be confounded by intra-patient heterogeneity[23]. An examination of lesion-specific responses to PD-1 inhibition revealed that 16/68 (24%) patients had heterogeneous tumor responses (Supplementary Data 1). Of these 16 patients, five had PRE core biopsies, allowing lesion-specific assessment of the response. Interestingly, four patients who had irRC PD underwent CR, PR, PR and SD in the lesion biopsied at baseline, and the one patient who had irRC PR underwent PD in the lesion biopsied (Table 2, Fig. 1D/E). The predictive accuracy of the seven anti-PD-1 predictive transcriptome signatures did not improve, however, when these lesion-specific responses were included in the ROC analyses (Supplementary Fig. 2A).

**Transcriptome evolution during PD-1 inhibitor progression.** We extended the transcriptome analysis to all patients ($n = 55$) and tumors ($n = 79$) with RNA sequencing data, including 44 PRE, 6 RES and 29 PROG melanomas (11 acquired, 18 innate). We initially examined the intra-tumoral cytolytic activity (CYT) score of each tumor, a quantitative measure of immune cytolytic activity based on transcript levels of two cytolytic effectors, granzyme A (GZMA) and perforin (PRF1)[24]. As expected, the CYT score correlated with computational estimates of immune cell fractions and immune activation signatures (Supplementary Fig. 2B, 2C). The CYT score was highest, but not exclusively elevated, in the RES biopsies (Fig. 2A). Fifteen of 44 (34%) PRE and 15/29 (52%) PROG tumors (6/11 acquired and 9/18 innate) showed evidence of an active inflamed transcriptome (i.e CYT score within the CYT score range of the 6 RES tumors) (Fig. 2A).

The patterns and evolution of tumor inflammation were explored in eight patients who had paired PRE and PROG tumors (patients 21, 24, 31, 44, 46, 47, 48, 49) (Fig. 2B). We noted discordance between PRE and PROG, or between multiple

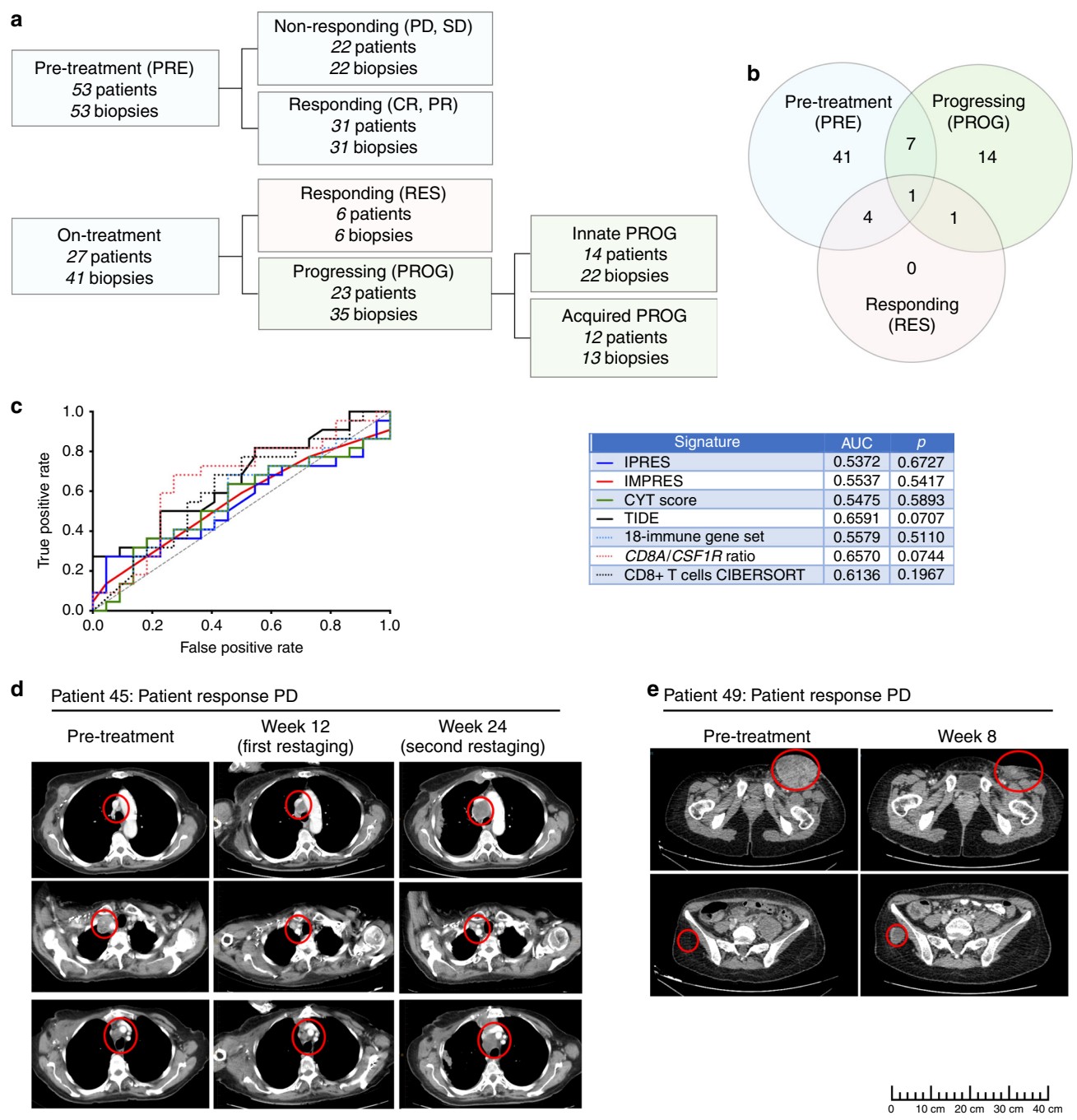

**Fig. 1 Predictive performance of transcriptome signatures and PD-1 inhibitor response heterogeneity. a** Details of the 94 melanoma biopsies analyzed in this study, including 53 pre-treatment (PRE) and 41 on-treatment samples. Of the 53 pre-treatment tumor specimens analyzed, 31 were obtained from patients who subsequently underwent complete response (CR) or partial response (PR) by irRC criteria and 22 pre-treatment biopsies were obtained from patients who had stable disease (SD) or progressive disease (PD) by irRC. **b** Venn diagram showing the 68 patients included in this study and the distribution of pre-treatment, responding and progressing tumor specimens ($n = 94$). **c** Immune-predictive transcriptome scores derived for each PRE-treatment melanoma biopsy ($n = 44$) and patient response data were used to generate receiver operator characteristic (ROC) curves measuring the performance of each indicated signature in predicting PD-1 inhibitor responses in our patient cohort. The resulting AUCs and p values are tabulated. The signatures applied to our dataset were derived from the following references: IPRES signature[17], IMPRES signature[21], *CD8A/CSF1R* ratio[15], 18-immune gene set[18], TIDE[22], CYT score[24] and CIBERSORT estimated relative proportion of CD8+ T cells[74] (see Supplementary Data 6). **d** CT scans from patient 45. Tumor metastases pre-treatment and on PD-1 inhibitor therapy (week 12 and 24) measured by CT images are shown. Regions of interest in CT images are circled in red. Top images show new lesion at week 12 that continued growing in size at week 24. Middle images show core biopsied lesion that underwent partial response. Lower images show pre-existing lesion that initially responded at week 12 but progressed by week 24. **e** CT scans from patient 49. Regions of interest in CT images are circled in red, and show partial response of large, inflamed pre-treatment inguinal LN metastasis (upper images) and the appearance of a new, subcutaneous buttock metastasis on treatment (week 8; lower images). Despite excision of the new metastasis, there were multiple new metastases in bone and lymph node on second restaging. Scale bar is shown.

**Table 1 Baseline clinicopathologic characteristics of melanoma patients.**

| Characteristics | Patients ($n = 68$) |
|---|---|
| Age, median (range) | 67 (38–88) |
| Sex, n (%) | |
| Male | 38 (56) |
| Female | 30 (44) |
| Prior BRAF±MEK inhibitor therapy | |
| Yes | 23 (34) |
| No | 45 (66) |
| M Stage (AJCC 8th edition), n (%) | |
| M1a | 6 (9) |
| M1b | 8 (12) |
| M1c | 38 (56) |
| M1d | 16 (23) |
| Mutation[a], n (%) | |
| BRAF$^{V600}$ | 19 (28) |
| NRAS | 16 (24) |
| Other[b]/none | 33 (48) |
| LDH at baseline, n (%) | |
| ≤ULN | 40 (59) |
| >ULN | 28 (41) |
| Treatment, n (%) | |
| Pembrolizumab | 49 (72) |
| Nivolumab | 19 (28) |
| Timing of biopsy | |
| PRE only | 41 (60) |
| On-treatment only | 15 (22) |
| Pre- and on-treatment | 12 (18) |
| Response[c], n (%) | |
| CR | 15 (22) |
| PR | 22 (32) |
| SD/PD | 31 (46) |

AJCC, American Joint Committee on Cancer; LDH, lactate dehydrogenase; ULN, upper limit of normal; CR, complete response; PR, partial response; SD, stable disease; PD, progressive disease.
[a]One patient had both a BRAF (G469E) and an NRAS (Q61L/R/P) mutation
[b]Including BRAF non-V600 mutations
[c]Patients were stratified into response groups based on immune-related response criteria. Patients with CR or PR were classified as responders, while patients with SD and PD were classified as non-responders

**Table 2 Patients with PRE tumors available who had heterogeneous tumor responses to PD-1 inhibition.**

| Patient ID | Patient response[a] | Pre-treatment tumor and subsequent response to PD-1 inhibition[b] | | Patient status |
|---|---|---|---|---|
| | | Lesion core biopsy site | Lesion response | |
| 24 | PR | Axillary LN | PD | Alive |
| 34 | PD | Thigh SC | SD | Dead |
| 35 | PD | Thigh SC | CR | Dead |
| 45 | PD | Axillary LN | PR | Dead |
| 49 | PD | Groin LN | PR | Alive |

CR, complete response; PR, partial response; SD, stable disease; PD, progressive disease; LN, lymph node; SC, subcutaneous.
[a]Patient response was determined using immune-related response criteria
[b]Pre-treatment tumor was core biopsied and response of this same lesion to PD-1 inhibition was evaluated

PROGs within the same patient, in 4/8 patients (patients 24, 31, 46, 48). For example, the PRE and three innate PROG tumors from patient 31 were non-inflamed, whereas the fourth innate PROG tumor from this patient, collected at the same time as the other three PROG tumors, was inflamed (Fig. 2B). The remaining four patients (Fig. 2; patients 21, 44, 47, 49) had consistent elevated CYT scores in the PRE and PROG biopsies (Fig. 2B). Re-analyses of a separate series of longitudinal melanoma biopsies collected pre-treatment, early on therapy and on progression in patients undergoing treatment with sequential CTLA-4 and PD-1 inhibition confirmed intra-patient heterogeneity of tumor inflammation (Supplementary Fig. 2D)[19].

**HLA-ABC downregulation is associated with de-differentiation.** We reasoned that mechanisms of PD-1 inhibitor resistance reflect the degree of immune cell infiltration and thus, we explored melanoma transcriptome signaling relative to the level of T-cell activity as defined by the CYT score[24]. As expected, the CYT score was positively correlated with the expression of the MHC class I genes in the 79 melanoma biopsies with RNA sequence data (Pearson correlations of 0.61, 0.77, 0.74 and 0.74 for HLA-A, -B, -C and B2M). However, the correlation of *HLA-A* transcript expression with the CYT score was diminished in PD-1 progressing biopsies compared to the PRE and RES tumors (Fig. 3A, Supplementary Fig. 3A). We did not identify any expressed alterations in the *B2M*, *HLA-A*, *-B* or *-C* genes in the

transcriptome data from the 79 melanoma biopsies, and the mutations identified in other genes associated with antigen presentation and IFNγ signaling (i.e., *JAK1*, *JAK2*, *STAT1*, *STAT2*, *IFNGR1*, *IFNGR2*, *TAP1*, *TAP2)* were not exclusively identified in poor responders or PROG biopsies (Supplementary Data 3). Thus, in our patient cohort, T-cell activity as measured by the CYT score was not strongly associated with *HLA-A* transcript expression in PD-1 progressing tumors.

In order to examine mechanisms that may influence the expression of *HLA* genes independently of CYT score, we stratified tumors according to CYT score, and identified 19 CYT-matched tumor pairs (Supplementary Fig. 3B) with variable *HLA-A* transcript expression (Fig. 3B). Comparative gene expression profiling of these *HLA-A* low ($n = 19$) versus *HLA-A* high ($n = 19$) expressing tumors revealed 530 differentially expressed genes ($q < 0.01$) (Fig. 3C; Supplementary Data 4).

The *SNAI1* gene, which encodes the transcriptional regulator of E-cadherin and epithelial to mesenchymal transition (EMT) was the most highly upregulated gene ($\log_2 FC = 3.7$, $q < 0.001$) in *HLA-A*-low tumors (Figs. 3B, 3C). *SNAI1* transcript expression was positively correlated with transcriptome signatures indicative of EMT, stromal, endothelial and cancer associated fibroblasts and inversely correlated with *HLA-A* transcript expression in our cohort (Fig. 3D) and in the TCGA Skin Cutaneous Melanoma dataset (Supplementary Fig. 3C). Geneset enrichment analysis (GSEA PreRanked; Hallmark gene sets Molecular Signature Database (MSigDb) and stromal cell signatures[25,26]) also confirmed increased EMT, TGFß-signaling, fibroblast and endothelial cell signatures along with pan-fibroblast TGFß response signature[25] in the *HLA-A* low tumors in our data set (Fig. 3E, Supplementary Data 5).

The downregulation of *HLA-A* in these CYT-score matched tumors was not associated with diminished CD8+ T-cell content (Supplementary Fig. 4A), or alterations in the frequency of other leukocyte subsets (based on CIBERSORT profiling; Supplementary Fig. 4B and Supplementary Data 6). *HLA-A* downregulation was also not related to IFN-γ gene sets and although we detected downregulation of *B2M* in a number of *HLA-A* low tumors, this did not reach statistical significance (Supplementary Fig. 4C). It is also worth noting that PRE tumors with downregulated *HLA-A* transcript expression were not enriched in patients who had received MAPK inhibitor therapy (Supplementary Data 1) prior to anti PD-1 treatment (Fisher's exact test, $p > 0.99$). Interestingly, in the PROG tumor derived from patient 53 with a pre-existing, dysfunctional STAT1$^{S316L}$ mutation[27] (Supplementary Fig. 5), low transcript expression of *HLA-A* occurred in the absence of

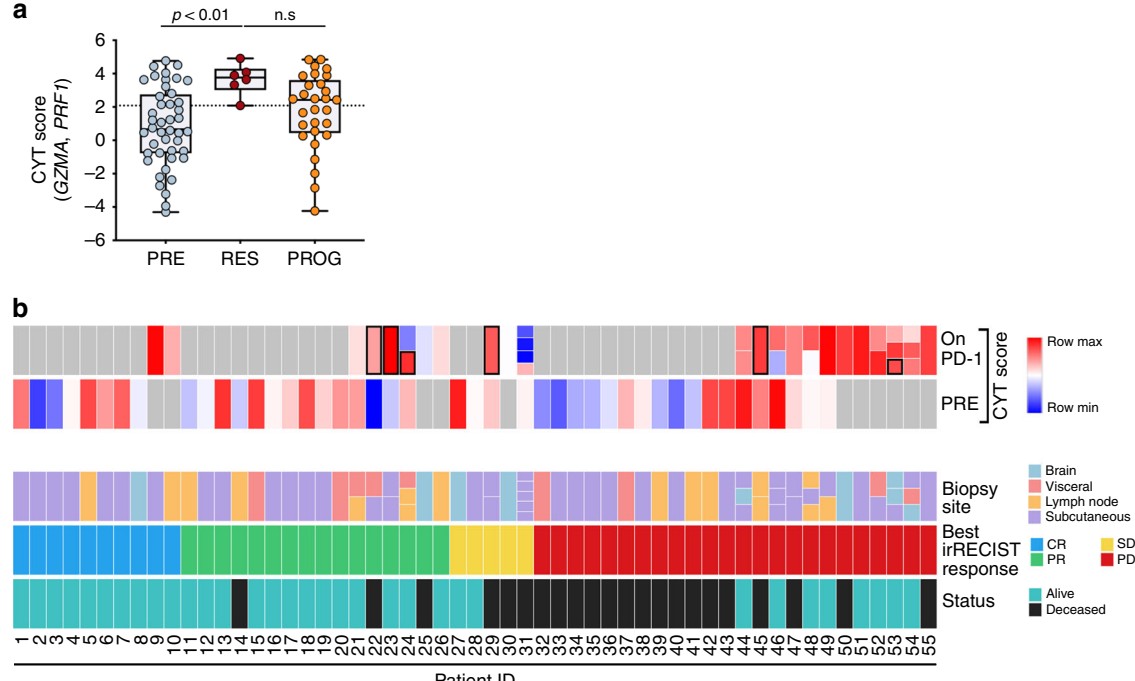

**Fig. 2 Immune cytolytic activity in longitudinal melanoma biopsies. a** CYT scores for the 44 PRE, 6 RES and 29 PROG melanoma biopsies. Box plots show the median and interquartile ranges, and CYT scores were compared using Kruskal-Wallis with Dunn's multiple comparison test. Dotted line aligns with the lowest RES tumor CYT score. n.s, not significant. **b** CYT score, clinicopathological and response details are shown for each of the 79 melanoma biopsies with RNA sequence data derived from 55 patients. The 6 responding on PD-1 inhibitor treatment biopsies are boxed. Where patients had multiple on treatment tumor biopsies these are shown as individual colored boxes within the 'On PD-1' and 'Biopsy site' rows.

elevated SNAI1 expression (Fig. 3). The second PROG tumor derived from patient 53 (also with the STAT1[316L] mutation, Supplementary Fig. 5) expressed similarly low levels of *HLA-A* (*HLA-A* $\log_2$ expression in Patient 53 PROG1 = 5.2, PROG2 = 4.5).

**HLA-ABC downregulation is common in melanoma biopsies.** To explore the frequency and contribution of HLA-A downregulation to immunotherapy resistance we analyzed 31 tumor dissociates derived from PD-1 PRE (*n* = 15, including 6 with RNA sequence data; Supplementary Data 1) and PD-1 PROG tumors (*n* = 16, including 10 from with RNA sequence data). Using flow cytometry, melanoma HLA-ABC expression was categorized as either normal or downregulated (ratio < 0.65) relative to the matching tumor infiltrating lymphocytes from the same biopsy sample. The immune cells served as an internal control that standardized the modulatory effects within the microenvironment. A representative gating strategy is shown in Supplementary Fig. 6. We observed varying degrees of cell surface HLA-ABC downregulation in 11/31 (35%) melanoma tumors (PRE 6/15 (40%) and PROG 5/16 (31%) samples) (Fig. 4A, B). Eight of these eleven tumors had an activating BRAF or NRAS mutation, and although BRAF[V600E] has been associated with the internalization of HLA-ABC from the cell surface[28], we did not detect any genotype-associated differences in the cell surface expression of HLA-ABC in 31 melanoma tumors (Supplementary Fig. 7A). It is worth noting that 16 melanoma PRE and PROG biopsies had matching flow cytometry and RNA sequence data and the HLA-ABC cell surface expression and HLA-A transcript expression were concordant in these samples (Spearman correlation 0.67, *p* < 0.01; Supplementary Fig. 7B). We also examined the cell surface expression of HLA-ABC and HLA-DR, which was recently shown to correlate with response to PD-1 inhibition[29], in the 15 PRE tumors. Although the tumor numbers were small, the

cell surface expression of HLA-ABC or HLA-DR at PRE did not accurately reflect patient response (Supplementary Fig. 7C & D).

The importance of HLA-ABC downregulation was confirmed using a short-term co-culture model of melanoma cells and autologous tumor infiltrating lymphocytes. Diminished expression of HLA-ABC (~80% reduction in expression; Fig. 4C) was driven by two independent B2M-targeting silencing molecules and this significantly reduced the immune recognition of melanoma cells, leading to approximately a 70% reduction in the levels of IFNγ production (Fig. 4C). As a comparison, pre-treatment of melanoma cells with an HLA-ABC blocking antibody reduced IFNγ secretion by immune cells by over 95% (Fig. 4D).

**HLA-ABC induction in PD-1 resistant melanoma.** We next examined the direct role of de-differentiation on HLA-ABC expression in a series of melanoma cell lines derived from PROG tumor dissociates. We noted that PD-1 PROG melanoma cell lines commonly expressed effectors and markers of de-differentiation[30–32], including accumulation of the receptor tyrosine kinase AXL[33] and downregulation of the micropthalmia-associated transcription factor (MITF)[34] (7/16; 44% PD-1 PROG cell lines) (Supplementary Figs. 8A, 9). Importantly, we confirmed that the melanoma de-differentiation phenotype was usually concordant between the tumor biopsy (defined by NGFR positivity) and the corresponding PD1 PROG cells models (Supplementary Fig. 8B). These MITF[low]/AXL[high] PROG cells demonstrated diminished HLA-ABC induction by exogenous IFNγ treatment (Fig. 5A). Furthermore, in our comparison of CYT score-matched *HLA-A* low (*n* = 19) vs *HLA-A* high (*n* = 19) tumors we observed markers of the MITF[low]/AXL[high] de-differentiated cell state, including downregulation of the *MITF/SOX10* regulated genes, *MITF, TYR, DCT* and *MLANA* transcripts (Supplementary Data 4) and upregulation of TGFß

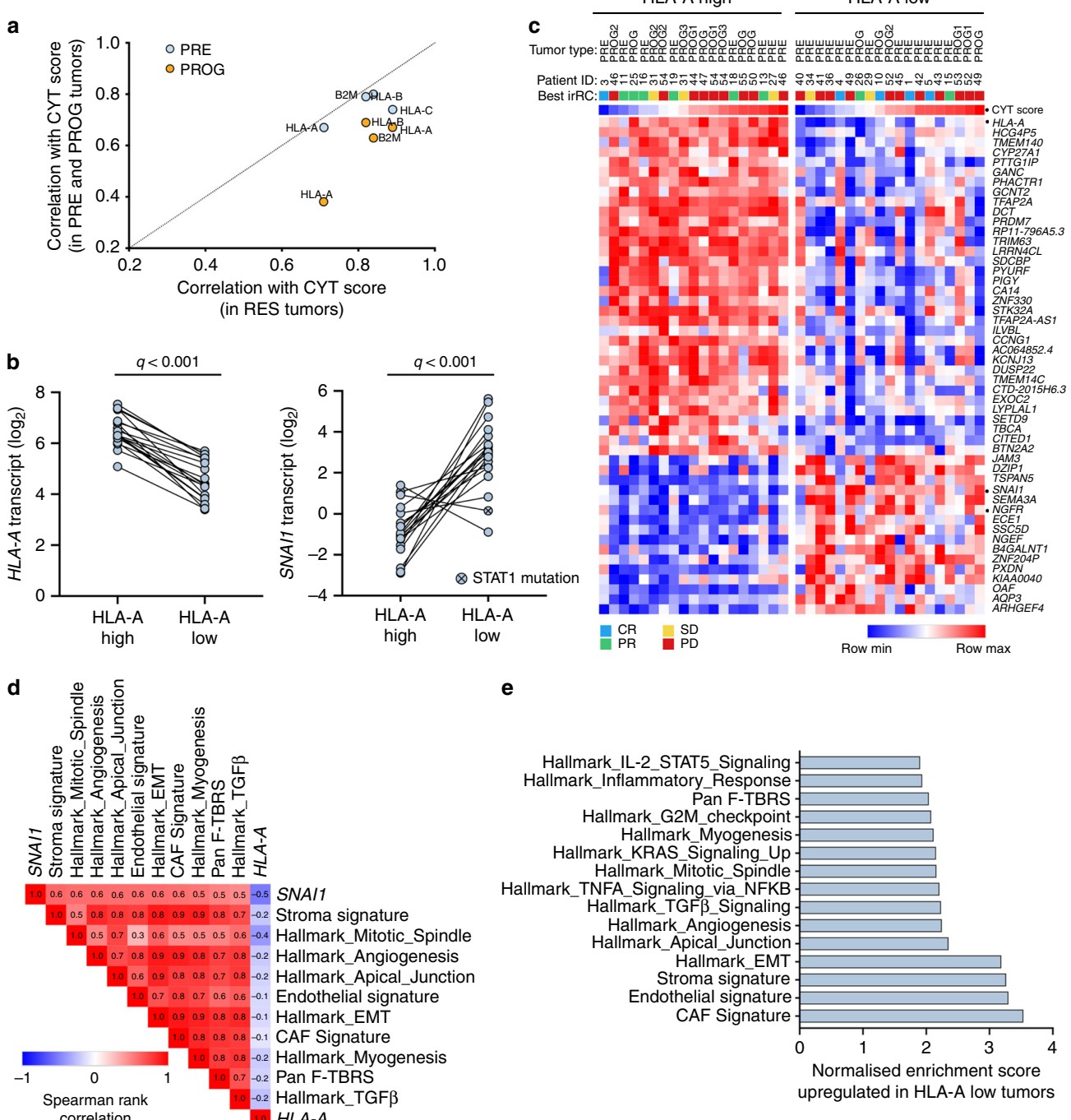

**Fig. 3 HLA-A transcript downregulation associated with markers of de-differentiation. a** Scatter plot showing the Pearson correlation coefficient of CYT score with the expression of MHC class I genes, *HLA-A*, *-B*, *-C* and *B2M* in responding (RES; $n = 6$), and pre-treatment (PRE; $n = 44$) and progressing biopsies (PROG; $n = 29$). **b** Plots showing expression of *HLA-A* and *SNAI1* in the CYT score-matched tumors ($n = 38$) with high or low *HLA-A* transcript expression. FDR-adjusted *p*-values (*q*) calculated using limma test. The *HLA-A* low melanoma tumor derived from patient 53 was found to express a STAT1$^{S316L}$ mutation (highlighted). **c** Heat map showing differentially expressed genes (FDR adjusted p-value < 0.001 are shown) between CYT score-matched tumors ($n = 38$) with low or high *HLA-A* transcript expression. CYT score is also shown and *HLA-A*, *SNAI1* and *NGFR* genes are highlighted. Best irRC response is also shown. **d** Correlation matrix of *SNAI1* gene expression with ssGSEA scores derived from the Hallmark gene set collection and stromal cell-specific transcriptome signatures[25,26] in 79 melanoma biopsies. The Spearman rank correlation coefficients are shown within the matrix, and the false discovery adjusted *p*-value was <0.01 for all signatures shown (see Supplementary Data 7). **e** Subset of top scoring genesets (GSEA PreRanked; Hallmark gene set collection and stromal cell-specific transcriptome signatures[25,26]) upregulated in the tumors with low *HLA-A* transcript expression compared to *HLA-A* high expressing tumors.

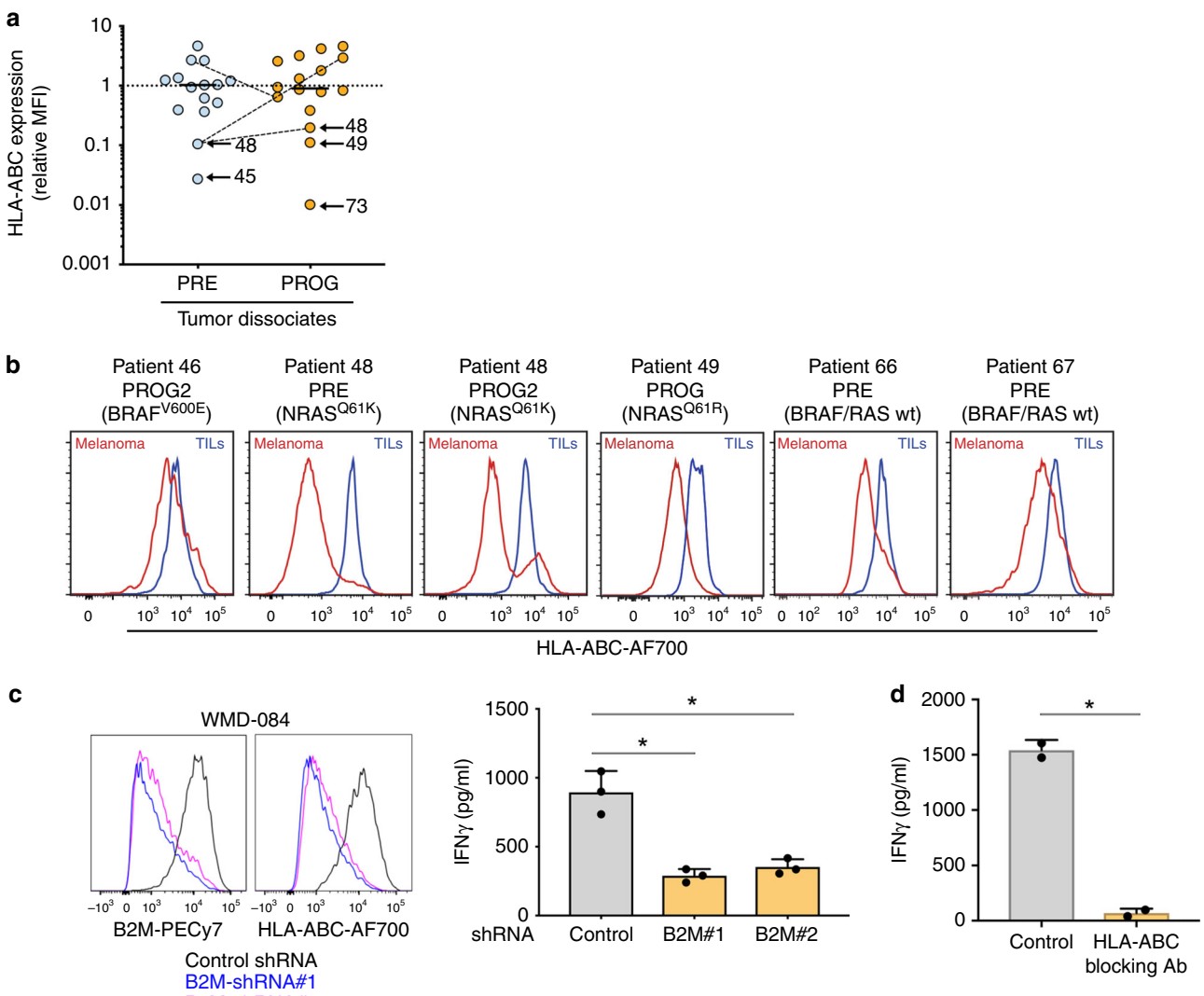

**Fig. 4 Immune checkpoint resistance in T-cell inflamed melanoma. a** Cell surface expression of HLA-ABC (relative to HLA-ABC in tumor-infiltrating lymphocytes) in melanoma cells from fresh dissociates of tumors derived pre-treatment (PRE) and progressing (PROG) on PD-1 inhibition. Solid line represents median and dotted line set at Y = 1. Patient-matched PRE and PROG tumors are connected with a dotted line. **b** Representative histograms showing levels of HLA-ABC expression in melanoma and tumor-infiltrating immune cells in PRE and PROG tumor dissociates. Red histograms show HLA-ABC expression in melanoma cells and blue histograms in tumor infiltrating tumors (TILs). Each tumor driver oncogene is also indicated. **c** Representative histograms showing cell surface expression levels of B2M and HLA-ABC in WMD-084 melanoma cells, 72 h post transduction with B2M-specific shRNA molecules (left panel). IFNγ production 72 h after co-culture of B2M silenced WMD-084 melanoma cells with the patient-matched tumor-infiltrating lymphocytes expanded from the same tumor biopsy. IFNγ was measured by ELISA (right panel). Data are means ± s.d. and individual data points represent the average of technical triplicates. Data were compared using one-way ANOVA with the Geisser-Greenhouse correction, *$p < 0.05$. **d** IFNγ production 72 h after co-culture of WMD-084 melanoma cells with the patient-matched TILs expanded from the same tumor biopsy. Lymphocytes were pre-treated for 1 h with 10 μg/ml HLA-ABC blocking or an isotype-matched antibody prior to co-culturing. Isotype antibody-treated control cells were compared to HLA-ABC blocking antibody-treated data using a paired *t*-test, *$p < 0.05$.

regulated genes, including *AXL* ($q = 0.06$), *TAGLN, NGFR, SERPINE1*, BGN transcripts (Supplementary Data 4).

The impact of TGFß on HLA-ABC expression and de-differentiation was next explored in two PD-1 PROG melanoma cells (WMD-084, SMU17-0132) and one PRE (SCC14-0257) melanoma cell line. Exogenous TGFß diminished cell surface expression of HLA-ABC at baseline and, in some instances, was sufficient to diminish IFNγ induced HLA-ABC in melanoma cell lines (Fig. 5B). TGFß exposure promoted variable levels of de-differentiation markers N-cadherin, AXL and SNAIL, with all three melanoma cell lines tested showing TGF-ß mediated induction of at least one of these markers (Fig. 5C & Supplementary Fig. 10). Moreover, TGFß reduced the recognition

of melanoma cells by autologous immune cells resulting in reduced IFNγ production (Fig. 5D).

## Discussion

This study confirms that MHC class I downregulation associated with the de-differentiation phenotype is a hallmark of both innate and acquired resistance to PD-1 inhibitors. Despite recent advances in the treatment of melanoma with immune checkpoint inhibitors, innate and acquired resistance remains a major challenge, with only modest improvements observed with second-line salvage therapies[35,36]. Genetic alterations in antigen presentation (B2M, HLA-A) and IFNγ signaling (IFNGR, STAT1 and JAK1/2) in non-responding melanoma patients[9,11–13] are uncommon, and

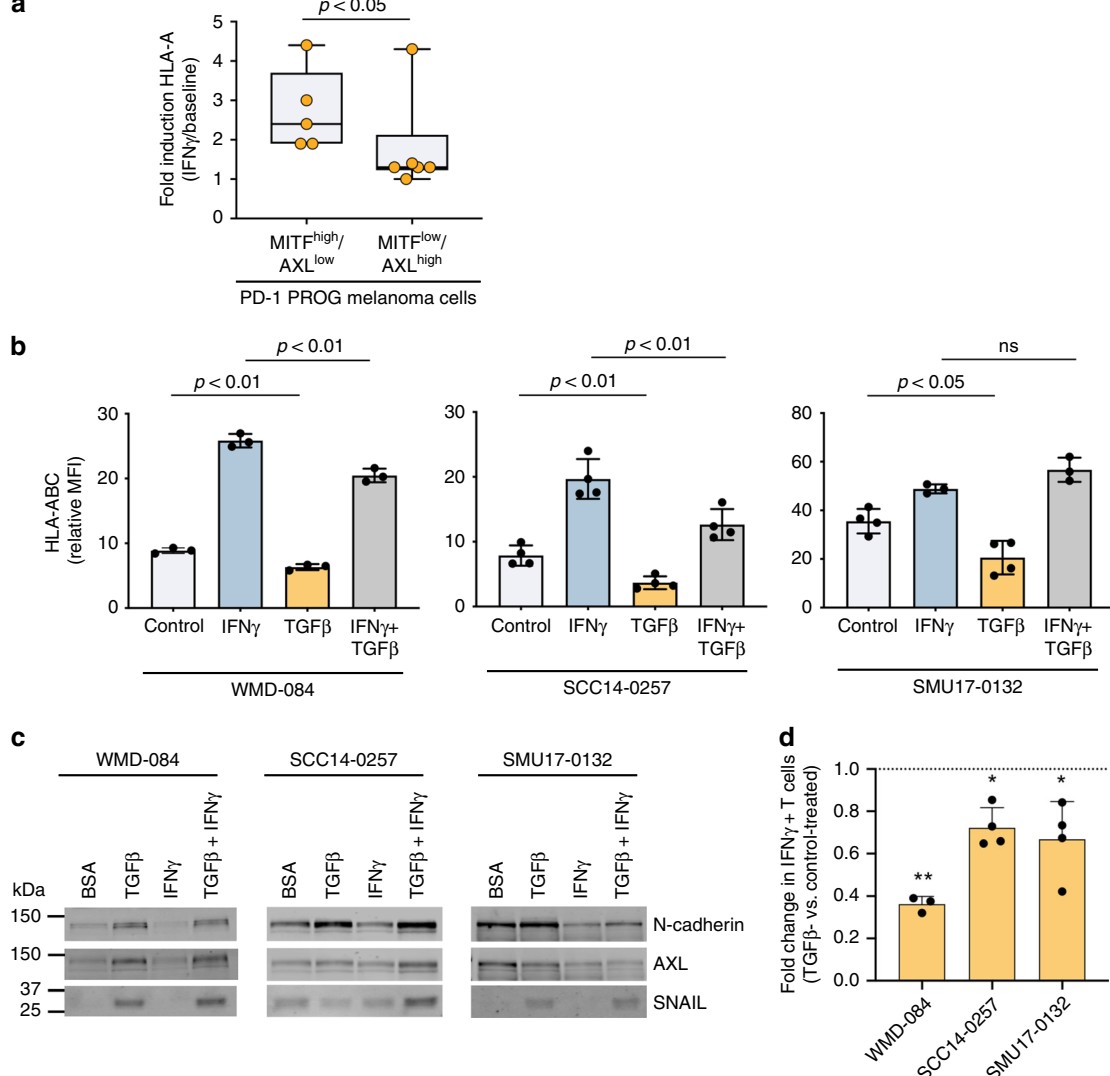

**Fig. 5 TGFß promotes HLA-ABC downregulation at baseline and in response to IFNγ. a** IFNγ-mediated induction (IFNγ-treated/vehicle-treated control) of cell surface HLA-ABC in MITF$^{high}$/AXL$^{low}$ or /MITF$^{low}$/AXL$^{high}$ short-term PD-1 PROG melanoma cell lines. Each dot represents one cell line and HLA-ABC induction was measured by flow cytometry 24 h after treating cultures with vehicle control or 1000 U/ml IFNγ. Box plots show the median and interquartile ranges, and data were compared using Mann-Whitney test. **b** Cell surface expression (median fluorescence intensity; MFI) of HLA-ABC in WMD-084, SCC14-0257 and SMU17-0132 melanoma cells treated with vehicle (Control), 1000 U/ml IFNγ and/or 10 ng/ml TGFß for 72 h. Data (mean ± s.d.) were compared using one-way ANOVA with the Geisser-Greenhouse correction. **c** Expression of de-differentiation markers AXL, N-cadherin and SNAIL in WMD-084, SCC14-0257 and SMU17-0132 melanoma cells treated with vehicle (Control), 1000 U/ml IFNγ- and/or 10 ng/ml TGFß for 72 h. **d** IFNγ production after co-culture of TGFß pre-treated (10 ng/ml for 72 h) melanoma cells with the patient-matched tumor-infiltrating lymphocytes expanded from the same tumor biopsy. IFNγ was measured by flow cytometry. Data (mean ± s.d.) show relative IFNγ expression in T cells (TGFß pre-treated/BSA pre-treated) after background subtraction (spontaneous IFNγ production on immune cell-only cultures). Paired BSA-treated vs TGFß-treated data were compared using paired t-test, **$p < 0.01$, *$p < 0.05$.

not restricted to non-responding patients[13,37,38]. We have now shown that MHC Class I downregulation occurred in 31% of PROG tumors, regardless of whether resistance was innate or acquired, with downregulation of MHC class I associated with TGFß activity, SNAI1 upregulation, cancer-associated fibroblast signatures and the MITF$^{low}$/AXL$^{high}$ melanoma phenotype. This has important implications when selecting subsequent combination immunotherapies in patients who have failed single agent anti-PD-1, as salvage strategies that depend on T-cell mediated anti-tumor immunity may not be effective.

The AXL$^{high}$ de-differentiation melanoma program has been linked to melanoma invasiveness[34], BRAF and MEK inhibitor resistance[39–41] and is associated with intrinsic resistance to PD-1 inhibitor monotherapy in melanoma patients[17]. AXL$^{high}$ melanoma are enriched during BRAF/MEK and PD-1 inhibitor therapy[41] and are induced via microenvironmental cues including T-cell-induced inflammatory stimuli (e.g., TNFα)[42,43] and cancer-associated fibroblast activity (e.g., TGFß signaling)[41,44,45]. The MAPK inhibitor and immune resistance effectors in de-differentiated melanoma may be distinct, however. In particular, the upregulation of multiple receptor tyrosine kinases, including AXL, PDGFRß and EGFR drive MAPK inhibitor resistance[39] whereas the global downregulation of melanocytic antigens[42,43] and the diminished upregulation of MHC class I expression (this report) mediate immune evasion.

Consistent with our data, AXL expression diminished STAT1 phosphorylation and MHC class I expression in a mouse mammary tumor model[46] and induced mesenchymal transition via TGFß/SNAI1-signaling, leading to the downregulation of MHC class I expression in prostate cancer[47]. Although, tumor cells lacking normal expression of MHC class I molecules should activate and be cleared by natural killer cells, the activation and function of these immune cells are likely impaired by the presence of stromal TGFß[48,49].

Recent data have also confirmed that fibroblast-derived TGFß restrained anti-urothelial cancer cell immunity and PD-1 response by restricting T-cell movement within the microenvironment[25] and elevated TGFβ1 diminished the positive effect of T-cell infiltration on melanoma patient survival outcomes[22]. Further, PD-1/PD-L1 blockade invoked limited response in preclinical colon cancer and mammary carcinoma mouse models with TGFß-activated stroma[25,50]. The combined inhibition of PD-1/PD-L1 with TGFß receptor kinase inhibitors has also been shown to enhance tumor regression in preclinical tumor models[51,52] and a bifunctional fusion protein targeting PD-L1 and TGFß in solid tumors showed encouraging efficacy in a recent Phase I trial[53]. Collectively, these data confirm that environmental TGFß cues contribute to immune evasion and PD-1 inhibitor escape by limiting T-cell infiltration and downregulating MHC class I expression.

In our large patient cohort of carefully annotated PROG samples, biopsied at time of progression, from a confirmed progressing metastases, we found that previously described alterations in antigen presentation (B2M, HLA-A loss) and IFNγ signaling (IFNGR, STAT1 and JAK1/2 alterations) were rare mechanisms of treatment failure[9,11–13]. None of our PROG tumors expressed B2M, IFNGR1, JAK1 or JAK2 mutations, and although we identified 5 PROG tumors, derived from three patents, with STAT1 mutations (patient 25—STAT1$^{S735F}$, patient 40—STAT1$^{S316L}$ and patient 50—STAT1$^{P633L}$), only patient 40 displayed concurrent STAT1 mutation and HLA-A downregulation.

Finally, we were unable to define a baseline transcriptome signature that accurately classified clinical response to PD-1 inhibitors nor could we confirm the predictive value of previously reported gene expression signatures[11,15,17,18]. Several recent reports have also been unable to validate published predictors of immunotherapy response. For instance, the IPRES gene expression signature did not accurately predict PD-1 inhibitor response in several independent melanoma cohorts[11,19,21]. CIBERSORT, cytolytic activity and IFN-γ gene expression signatures were poor predictors of PD-1 and CTLA-4 inhibitor response across several publicly available patient datasets[21,54] and the validity and reproducibility of the IMPRES gene expression signature in predicting immune checkpoint inhibitor response remains contentious[55,56].

Although baseline predictive information is eagerly sought to guide the selection of first-line immune therapies, our findings suggest that a single-point, robust pre-treatment biomarker is unlikely to represent the heterogeneous nature of cancer, or predict the rapidly evolving profile of tumors under the selective pressure of immunotherapies. The longitudinal analyses of multiple pre-treatment biopsies derived from various sites will be required to validate the influence of tumor heterogeneity on clinical outcomes and on the accuracy of baseline predictive signatures. In this study we observed significant variation in the response of individual lesions to immune checkpoint inhibition within melanoma patients, with heterogeneity of tumor responses observed in 85% and 69% of melanoma patients who did not achieve an objective response to pembrolizumab monotherapy or combination ipilimumab plus nivolumab, respectively[23,57]. This heterogeneity presumably reflects genetic, molecular and cellular variables, which may include the presence of neoantigens in only a subset of tumor clones, the expression of PD-L1 in a small proportion of tumor cells, variable expression of PD-1 on immune cells and intra-tumoral differences in T-cell density and clonality[58–60]. It will be interesting to explore whether these limitations influence the predictive value of tumor mutation load, which has been shown to predict clinical benefit of immune checkpoint blockade in multiple cancers, including melanoma[61–63]. Recent data suggest that tumor mutation burden may not change significantly during treatment[64], although intra-patient mutation burden heterogeneity has been observed[65]. Further, when tumor samples were stratified according to melanoma subtype (i.e., cutaneous, occult, acral or mucosal), tumor mutation burden did not predict response to anti-PD-1 based immune checkpoint therapies[66]. Unfortunately, germline variant data was not available in this study and it was not possible to accurately estimate tumor mutation burden in this study,

We conclude that MHC class I downregulation associated with the de-differentiation phenotype is a common mechanism of resistance to PD-1 inhibitors. With the availability of synchronous and longitudinally collected biopsy samples from melanoma patients, we were able to address the issue of immune and intra-patient heterogeneity as a significant limiting factor in identifying predictive signatures and in devising strategies to overcome immune checkpoint inhibitor resistance. Nevertheless, combination immunotherapy strategies such as the addition of CTLA-4 inhibitors that utilize additional anti-immunity pathways without solely depending on T-cell mediated anti-tumor immunity, and mechanisms that restore antigen presentation by inhibiting TGFß signaling, may improve the outcomes of patients with metastatic melanoma who are progressing on PD-1 inhibitors.

## Methods

**Patient, response assessment and tumor biopsies.** This study included 68 metastatic melanoma patients who were treated with PD-1 inhibitors (pembrolizumab or nivolumab) at Melanoma Institute Australia (MIA) and affiliated hospitals. Written consent was obtained from all patients (Human Research ethics committee protocols from Royal Prince Alfred Hospital; Protocol X15-0454 & HREC/11/RPAH/444). Tumor response was assessed using the immune-related response criteria (irRC[20]), with heterogeneous response defined as the presence of progressing and/or new metastases in conjunction with at least one responding metastasis on first restaging imaging. Clinicopathological characteristics including American Joint Committee on Cancer (AJCC) stage, lactate dehydrogenase (LDH) and mutation status were recorded (Supplementary Data 1), and follow-up duration was calculated from the date of first dose of systemic therapy to the following three dates: date of death, loss to follow-up or 30$^{th}$ November 2018.

**RNA isolation and sequencing.** Total RNA was isolated from 79 fresh frozen tissue sections using the AllPrep DNA/RNA/miRNA Universal kit (Qiagen, Hilden, Germany)[16]. cDNA synthesis and library construction were performed using the TruSeq RNA Library Prep Kit (Illumina) and paired-end 100 bp sequencing, with each sample yielding 40–50 million read. Sequencing was performed on the Illumina Hiseq 2500 platforms at the Australian Genome Research Facility in Melbourne.

**RNA-sequence data processing.** Trimming of Illumina TruSeq paired-end sequencing adapter sequences and bases with a quality of <20 from each end was done using cutadapt[67]. Reads less than 50 bases long after trimming were discarded from subsequent analysis. The filtered reads were mapped to reference genome hg38 using STAR with 10% mismatches allowed. Reads that mapped equally to more than one genomic location were discarded. Reads were imported into R with GenomicAligments read GAlignmentPairs function. strandMode was set to 2. GENCODE Genes version 26 was used as the gene reference database. Counting of reads overlapping with exonic regions of each gene was done with the countOverlaps function from GenomicAligments.

**Differentially expressed gene analysis.** RNA count data were normalized using trimmed mean of M-values (TMM) and transformed with voom to log2-counts per million with associated precision weights[68]. To identify differentially expressed genes associated with variable levels of *HLA-A* transcript expression we compared all PRE and PROG tumor pairs for CYT score and selected tumor pairs with

similar CYT scores (ratio between 0.9–1.1). These were ranked according to *HLA-A* transcript differences and tumor pairs with *HLA-A* expression differences greater than two-fold were selected. Nineteen tumors pairs were categorized as low *HLA-A* transcript relative to CYT score and high *HLA-A* transcript relative to CYT score. Differentially expressed genes between these two groups were determined using the moderated t-tests (implemented in R package version 3.6.0 limma version 3.40.2) based on an empirical Bayesian approach to estimate gene expression changes[69]. Similarly, we grouped patients into responders (complete and partial response) and non-responders (stable and progressive disease) and also applied moderated t-test (implemented in limma) to identify gene expression associated with response.

**Gene set and cell type enrichment analysis**. Rank ordering of TMM-voom transformed gene expression data was carried out using the linear model for microarray module (limma package in R/Bioconductor)[69] and analyses was performed using gene set enrichment analysis in pre-ranked mode provided by GenePattern[70]. The Hallmark gene sets[71] of the Molecular Signature Database version 6.2[72], with stromal and IPRES cell gene signatures[17,25,26] were considered. A false discovery (FDR) corrected p-value <0.05 was used for comparisons between CYT-matched melanoma tumors.

To obtain abundance values corrected for transcript lengths as required by the single-sample gene set enrichment analysis (ssGSEA[73]), RSEM was used to derive the FPKM estimates using GENCODE Genes version 26 as the reference transcript database. Absolute signature enrichment scores were determined using the ssGSEA (version 9.1.1) implementation provided by GenePattern[70] with the gene sets described above. Subsequently, differential expression analyses on ssGSEA enrichment scores was performed using the moderated t-test (limma package in R/Bioconductor)[69]. A false discovery (FDR) adjusted p-value <0.05 was used for comparisons between tumor groups. The same FPKM values were also used to infer the relative proportions of 22 types of infiltrating immune cells using the CIBERSORT web portal (http://cibersort.stanford.edu/). These 22 cell subsets were further grouped into 11 major leukocyte subtypes[74] and a moderated t-test (implemented in limma) was applied to identify cell subsets associated with HLA-A transcript expression.

The correlation between transcript expression and ssGSEA enrichment scores was calculated using the Spearman's Rank correlation coefficient in the nearest neighbor algorithm within the Morpheus web based tool (https://software.broadinstitute.org/morpheus/).

The PD-1 predictive signatures applied to our dataset were as follows: IPRES signature (average Z scores of ssGSEA scores of the gene sets in the IPRES signatures[17], IMPRES signature[21], *CD8A/CSF1R* ratio (numeric difference between TMM-voom transformed *CD8A* and *CSF1R* expression data[15], 18-immune gene signature (ssGSEA score of 18-genes in the immune signature[18]) TIDE (calculated using TMM-voom sequencing counts normalized for each gene by subtracting the average gene values among all samples at http://tide.dfci.harvard.edu[22], CYT score (average of TMM-voom transformed *GZMA* and *PRF1* expression data[24]) and CIBERSORT estimated relative proportion of CD8+ T cells using FPKM expression estimates (http://cibersort.stanford.edu[74]). These predictive scores were derived for each pre-treatment melanoma biopsy ($n = 44$) and used with patient response data (complete (CR) and partial response (PR) versus stable (SD) and progressive disease (PD)) to generate receiver operator characteristic (ROC) curves in order to measure the performance of each indicated signature in predicting PD-1 inhibitor responses in our patient cohort. The performance of these seven PD-1 predictive signatures in predicting responding (irRC: CR and PR) were also evaluated in three publicly-available pre-treatment melanoma RNA-seq datasets with response data: (1) 49 patients treated with the PD-1 inhibitor, nivolumab[11], 26 patients treated PD-1 inhibitors nivolumab or pembrolizumab[17] and 41 patients treated with anti-CTLA4[75]. The area under the ROC curve was calculated using GraphPad version 8.2.1 using non-parametric estimates and 95% confidence interval based on the hybrid method of Wilson and Brown[76].

**Single nucleotide variant (SNV) analysis**. SNVs were called against the reference genome using VarScan2. Minimum variant frequency was set to 20% and other parameters were left at their default values. Briefly, the SAMtools mpileup utility provided a summary of the read coverage, and the mpileup output was processed using VarScan2 to call variants and produce a VCF format file with variants that passed the minimum read and allele frequency thresholds. Insertion and deletion calls were not included due to positional ambiguity and low alignment accuracy[77]. Visualization of the resulting VCF files and analysis was performed through the use of Ingenuity Variant Analysis software (https://www.qiagenbioinformatics.com/products/ingenuity-variant-analysis) from Qiagen.

**Tissue processing and cell isolation**. Tumor biopsies were manually minced and enzymatically processed, then dissociated into single-cell suspensions using the human Tumor Dissociation Kit and gentleMACS Dissociator (Miltenyi Biotec), according to the manufacturer's instructions.

Single-cell suspensions were viably frozen as tumor dissociates (TD, $1 \times 10^6$ cells/vial) in 10% DMSO in human serum from male AB plasma (Sigma) and

plated into 24 well plates ($1 \times 10^6$ cells/well) to isolate short term melanoma and tumor infiltrating lymphocyte (TIL) cultures.

**Cell culture**. Short term melanoma cultures were maintained in Dulbecco's Modified Eagle media supplemented with 10% heat inactivated fetal bovine serum (FBS; Sigma Aldrich, St. Louis, MO, USA), 4 mM glutamine (Gibco, Thermo Fisher Scientific, Waltham, MA, USA), and 20 mM HEPES (Gibco), at 37°C in 5% $CO_2$. Cell authentication and profiling of established and newly derived cell lines was confirmed using the StemElite ID system from Promega. All cells tested negative for mycoplasma (MycoAlert Mycoplasma Detection Kit, Lonza, Basel)

TILs were cultured in TIL media (Roswell Park Memorial Institute (RPMI) 1640 media supplemented with 10% heat inactivated human serum from male AB plasma (Sigma), 25 mM HEPES, 100 U/ml penicillin, 100 µg/ml streptomycin, 10 µg/ml gentamycin, 2 mM L-glutamine and 1000 U/mL IL-2 (Peprotech) and expanded with addition of Dynabeads Human T activator CD3/CD28 (ThermoFisher, 25 µl/ml media).

For IFNγ and TGFβ treatment, $6 \times 10^5$ melanoma cells were plated in T75 cm flasks. After an overnight incubation, media was replenished, and cells treated for 72 h with 1000 U/ml IFNγ (Peprotech, Rocky Hill, NJ, USA), 10 ng/ml TGFβ (Peprotech, Rocky Hill, NJ, USA), combination of both, or vehicle control (0.1% bovine serum albumin (Sigma-Aldrich) in phosphate-buffered saline (PBS, Gibco)). Cells were collected, washed with PBS, and analyzed by flow cytometry and immunoblotting.

**Co-culture melanoma:TIL assays**. IFNγ production in melanoma:TIL co-cultures was analyzed by ELISA or flow cytometry. To measure IFNγ release, $1 \times 10^4$ melanoma cells were cultured with $1 \times 10^4$ TILs (1:1 effector to tumor ratio) in a 96-well plate in a total volume of 100 µl TIL medium, and each experimental setup was performed in triplicate. After two days culture, supernatant was collected, spun down to remove cell debris, and stored at −20 °C for IFNγ analysis using the Human IFNγ DuoSet ELISA (R&D Systems). ELISA was performed according to manufacturer's protocol. Alternatively, frequency of IFNγ-producing CD8+ T cells was measured by flow cytometry, For flow cytometry, $1 \times 10^5$ melanoma cells were cultured with $1 \times 10^5$ autologous TILs in a 24-well plate in 0.5 ml of TIL medium. Four hours post co-culture, 5 µg/ml of Brefeldin A and 5 µg/ml monensin (both from Sigma) were added to stop cytokine release and the co-cultures were incubated overnight prior to staining (see below). For TGFβ treatment, melanoma cells were pre-treated with 10 ng/ml TGFβ or vehicle control for 72 h prior to co-culture with TILs.

For B2M silencing experiments, melanoma cells were first transduced with negative control shRNA or B2M shRNA before co-culturing with autologous TILs, as described above. HLA-ABC blocking was performed using a monoclonal mouse anti-HLA-ABC antibody (clone W6/32, Cat No. 311409, Biolegend). Melanoma cells were pre-treated with 20 µg/ml anti-HLA-ABC antibody or mouse IgG2a isotype control antibody (Biolegend) for 1 before co-culture with autologous TILs.

**Constructs and lentivirus transductions**. The B2M shRNA constructs correspond to nucleotides 144–162 and 402–420 (Genebank accession number NM_004048.2)[78]. The non-silencing negative control shRNA did not show complete homology to any known human transcript and had the following sequence: 5'-TTAGAGGCGAGCAAGACTA-3'. The shRNA were cloned into *pSIH-H1-puro* (System Biosciences) lentiviral vector. Lentiviruses were produced in HEK293T cells as described previously[79]. Cells were infected using a multiplicity of infection of 5 to provide an efficiency of infection above 90%. Cells were used 72 h post transduction.

**Immunoblotting**. Total cellular proteins were extracted at 4°C using RIPA lysis buffer containing protease inhibitors and phosphatase inhibitors (Roche). Proteins (15–40 µg) were resolved on 8-12% SDS-polyacrylamide gels and transferred to Immobilon-FL membranes (Millipore). Western blots were probed with antibodies MLANA (1:1000; Cell Signalling Technology; Cat. No. 34511), SOX10 (1:1000; D5V9L; Cell Signalling Technology; Cat No. 89356), AXL (1:200; R&D System; Cat No. AF154), MITF (1:1000; C5; Calbiochem; Cat. No. OP126L), N-cadherin (1:2000; 3B9; Invitrogen; Cat No. 33-3900), SNAIL (1:1000; C15D3; Cell Signalling Technology; Cat No. 3879) and ß-Actin (1:6000; AC-74; Sigma-Aldrich; Cat No. A5316). Where indicated, membranes were incubated with REVERT 700 total protein stain (LI-COR, Lincoln, NE), imaged using Odyssey CLx imaging system, washed and blocked using LI-COR Odyssey blocking buffer.

Protein expression data for MITF was normalized to ß-actin and N-cadherin, AXL, MLANA, SOX10, SNAIL were normalized to the REVERT 700 total protein stain (LI-COR). Z-scores for each protein were calculated from the normalized expression data derived from each independent biological replicate.

**Flow cytometry**. Staining was performed in flow cytometry buffer (PBS supplemented with 5% FBS, 10 mM EDTA, and 0.05% sodium azide). Cells ($2 \times 10^5$) were incubated for 30 min on ice with mouse anti-human antibodies against HLA-ABC (clone W6/32; BioLegend, San Diego, CA) conjugated to phycoerythrin (PE) (1:100; Cat. No. 311406) or Alexa Fluor 700 (1:80; Cat. No. 311438); HLA-DR,DP, DQ (clone Tu39; BD Biosciences) conjugated to Brilliant Ultraviolet (BUV) 395

(1:200; Cat. No. 740302), CD45 (clone HI30; BioLegend) conjugated to Brilliant Violent (BV)711 (1:200; Cat. No. 304050); SOX10 (clone A2; Santa Cruz Biotechnology, Santa Cruz, CA) conjugated to Alexa Fluor (AF) 488 (1:50; Cat. No. sc-365692), CD271/nerve growth factor receptor (clone ME20.4; BioLegend) conjugated to PE-cyanine (Cy)7 (PE-Cy7) (1:100; Cat. No. 345110) or beta-2-microglobulin (clone 2M2; BioLegend) conjugated to PE-Cy7 (1:200; Cat. No. 316318). Fc block (1:200; BD Biosciences; Cat. No. 564220) was used to prevent non-specific staining due to antibody binding to Fc receptors. Prior to acquisition, cell viability was determined by staining cells with either 5 μM 4'6-diamidino-2-phenylindole (DAPI) (Invitrogen, Thermo Fisher Scientific), or Live Dead near-infrared (NIR) fixable dye (Invitrogen, Thermo Fisher Scientific). Frequency of IFNγ producing cells in melanoma:TIL co-cultures was analyzed using intracellular staining. Briefly, cells were collected and stained with fixable Live Dead NIR (ThermoFisher) and Fc block, followed by antibodies against CD3 (1:100; clone UCHT1; BD Biosciences, Cat. No. 565491), CD8 (1:100; clone SK1; BD Biosciences; Cat. No. 561617) and CD4 (1:100; clone SK3; BioLegend; Cat. No. 344604), conjugated to BV786, V500 and Fluorescein isothiocyanate (FITC), respectively. Cells were fixed and permeabilized using the BD Biosciences Cytofix/Cytoperm Fixation/Permeabilization kit, stained with AF647-conjugated anti-IFNγ antibody (1:20; clone 4 S.B3, BD Biosciences, Cat. No. 563495) plus Fc block in permeabilization buffer, extensively washed and immediately analyzed.

All samples were acquired on BD LSRFortessa X20 flow cytometer (BD Biosciences) and analyzed using FlowJo software v10.4 or later (TreeStar, Ashland, OR). At least 10,000 live events were acquired, while all events were collected for the dissociated tumor samples. Relative marker expression levels were calculated by dividing the median fluorescence intensity (MFI) of the antibody-stained sample by the unstained control, unless otherwise specified. For dissociated tumor samples, melanoma HLA-ABC expression was calculated relative to TILs (geometric mean fluorescence intensity, MFI HLA-ABC melanoma/MFI HLA-ABC TILs).

**Statistical analyses**. For statistical analysis, we used GraphPad Prism software v8.1.1. Figure legends specify the statistical analysis used and define error bars.

## Data availability

TCGA gene expression data was downloaded from the The National Cancer Institute (NCI) Genomic Data Commons (GDC) database using the R/Bioconductor package 'TCGAbiolinks'63. BROAD javaGSEA standalone version can be downloaded from http://www.broadinstitute.org/ gsea/downloads.jsp. The RNAseq data is deposited in the European Genome-phenome Archive (EGA) under study accession number EGAS00001001552, and dataset accession EGAD00001005738. All other data is available within the Article, Supplementary Information or available from the authors upon reasonable request.

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

## Acknowledgements

This work was supported by Melanoma Institute Australia, the New South Wales Department of Health, NSW Health Pathology, National Health and Medical Research Council of Australia (NHMRC), Cancer Institute NSW and the Sydney Vital Translational Cancer Research Centre. J.H.Y.Y., J.S.W., H.R., R.A.S., and G.V.L. are supported by NHMRC Fellowships. G.V.L. is also supported by the University of Maryland Foundation. A.M.M. is supported by a Cancer Institute NSW Fellowship. J.F.T. is supported by the Melanoma Foundation of the University of Sydney. SYL is supported by grant #1123911, awarded through the Priority-driven Collaborative Cancer Research Scheme and co-funded by Cancer Australia and Cure Cancer Australia Foundation.

## Author contributions

H.R. and J.H.L. conceived the idea and supervised the study; E.S., S.Y.L., A.S., B.P., M.I. and S.A. performed the experiments; H.R., J.H.L, E.S., S.Y.M, A.S., B.P., M.I. and D.S. analyzed the data; H.R. and J.H.L. wrote the manuscript. All authors discussed the results and commented on the manuscript.

## Competing interests

J.H.L. has received honoraria from AstraZeneca and travel support from BMS. G.L. receives consultant service fees from Amgen, BMS, Array, Pierre Fabre, Novartis, MSD, and Roche. A.M.M. is on the advisory board of BMS, Merck (MSD), Novartis, Roche, and Pierre Fabre. R.F.K. has been on advisory boards for Roche, Amgen, BMS, MSD, Novartis and TEVA and has received honoraria from MSD, BMS and Novartis. M.C. is an advisory board member for MSD, BMS, Novartis and Amgen. The remaining authors declare no competing interests.
