## [Peer Review File · Nature Communications]

Reviewers' comments:

Reviewer #1 (Remarks to the Author):

Summary:

In the manuscript "Transcriptional downregulation of MHC class I and mesenchymal transition in melanoma resistance to PD-1 inhibition" the authors aim to study mechanisms of resistance to checkpoint inhibitor therapy (anti PD-1). For that purpose the authors have collected an impressive dataset of 65 melanoma patients (pre & on treatment) and performed transcriptome analysis of 44 patients (pre treatment) & 23 patients (35 samples) (on treatment). In addition, matching flow cytometry data was available for a subset of patients. The study provides a great resource for the study of resistance mechanisms and is of great value for the research community (provided the authors release their data).

First, the authors demonstrated that existing immune-related transcriptomic signatures fail to accurately distinguish responders from non-responders in their dataset. They argue that intra-patient heterogeneity could limit the predictive performance, however, they observed it in only 4 patients of their own dataset.

Next, they aimed to study mechanisms of resistance. They observed a diminished correlation of CYT score with HLA-A gene expression which led to a deeper investigation of 'non-genetic' mechanisms of HLA-A gene expression. To identify the genes that are associated with HLA-A underexpression independent of CYT score, the authors performed differential expression analysis between HLA-low and HLA-high expressing tumors, while controlling for CYT score. This analysis revealed several differentially expressed genes, enriched with gene set signatures of EMT, TGF- β signaling, fibroblast and endothelial cell. The authors confirm with flow cytometry HLA-ABC cell surface expression and in co-culture assays that TGF- β drives downregulation of MHC-class I. The authors conclude that MHC-I downregulation associated with a 'de-differentiation' gene set signature is a common mechanism of resistance to anti PD-1 therapy.

The question the authors address is important, however, we fail to understand the rationale of the study and we think the data presented are not sufficient to support the main claims.

Major comments:

1. The research question and hypothesis of the study is not clear. Is it MHC-I down-regulation or de-differentiation marker that is responsible for anti-PD1 resistance? In the first sentence of Discussion, the authors state "This study confirms that MHC class I downregulation associated with the de-differentiation phenotype is a hallmark of both innate and acquired resistance to PD-1 inhibitors." We think the authors totally failed to provide sufficient support for this statement. If the de-differentiation phenotype or downregulation of MHC-I is the hallmark of both innate and acquired resistance to PD-1, please demonstrate that de-differentiation markers are sufficient to distinguish, at least, responders vs "nonresponders with innate resistance" among PRE-samples, or something comparable to that.

2. In continuation of comment 1, we fail to understand this statement "Not surprisingly, cell surface expression of HLA-A in the 12 PRE tumors did not reflect patient response, although 2 of 4 non-responding PRE tumors showed HLA-A downregulation (Patient 45 and 48; Figure 4A)" Why the authors study the down-regulation of MHC-I as mechanism for resistance to anti-PD1 while their low expression is not expected in non-responders? If down-regulation of HLA is such a determining factor, this should be reflected in the dataset that the authors analyze. Please demonstrate that the down-regulation of HLA is associated with better response in your dataset.

3. The authors compare several predictive scores/signatures of response to checkpoint inhibitors, but it is not clear how the AUROC was calculated. In particular, (i) how the positive and negative sets were determined, (ii) how the authors resolve the cases where one patient has multiple samples, (iii) how different signature values were calculated. We suggest the following to resolve

the issue. (1) The methods to calculate the AUROC should be clearly stated. (2) We propose to present all signature values and positive/negative labels for each sample in your dataset as Supplementary Table.

4. In addition, the authors claim that "For example, the innate PD-1 inhibitor resistance (IPRES) signature, which includes 26 genes associated with mesenchymal transition and BRAF/MEK inhibitor resistance, was associated with lack of PD-1 inhibitor response in pre-treatment melanoma biopsies in one study, but was not associated with PD-1 inhibitor response in other melanoma cohorts." We agree that the validation of IPRES signature is limited, but we know other signatures such as IMPRES was demonstrated to be predictive in about ten independent melanoma datasets. Thus, we suggest to calculate AUROC with randomly picking up only one sample per patient in a given iteration, repeat this process for all possible configurations, and present the distribution of AUROC (mean and standard deviation).

5. The result that the authors present may be confounded with driver mutations. Bradley et al. (2015, Cancer Immunology Research) showed in melanoma cancer cell line model that BRAF-V600E mutations promote MHC-I internalization and re-location to the endolysosomal compartments. In Figure 4A, 4B cell surface expression are shown for patients with available flow cytometry data. Interestingly, patients 64 with the lowest HLA-ABC cell surface expression, but also patient 46 had a driver mutation in BRAF-V600E. Other patients with downregulated surface HLA-ABC had other mutations in MAPK pathway (48-NRAS Q61R, 49-NRAS Q61R, 45 -BRAF-V600K, 49-NRAS Q61R. This observation points to the possibility of other mechanisms (eg. non TGF-beta but mutation driven) of MHC-I surface downregulation Could the authors include as a control, patients with low HLA-ABC surface expression and no mutations in genes involved in the MAPK pathway?

6. The authors write in the introduction section 'we demonstrate that non-genomic down-regulation of HLA class I expression, rather than complete loss of HLA class I molecules..' and discussion section 'We have now shown that non-genetic MHC class I downregulation occurred in..' The methods section only describes variant calling for single nucleotide variants (SNVs) and does not include insertions and deletions that could lead to potential frameshifts mutations (eg. Zaretsky et al, 2016). Therefore, it is speculative to assume all HLA class I downregulation mechanisms are non-genomic or non-genetic. The authors should perform indel calling on their datasets and also discuss this limitation in the manuscript.

7. The authors explain in the methods section the estimation of the relative proportion of 22 types of infiltrating immune cells using CIBERSORT. In Figure S2C only CD8+ CIBERSORT extracted cells were compared between HLA-A(low) vs HLA-A(high). Could the authors perform a systematic analysis of the 22 types of immune cells and report the findings in a supplementary figure?

8. Many of the important data are not transparently presented. (1) Please show a scatter plot of CYT score and HLA-A expression in all tumors with different color-coding for PRE, RES, PROG tumors. (2) "It is worth noting that 16 melanoma PRE and PROG biopsies had matching flow cytometry and RNA sequence data and the HLA-ABC cell surface expression and HLA-A transcript expression were concordant in these samples (Spearman correlation 0.66, $p < 0.01$; data not shown)." We think this data should be presented. (3) The authors state on page 12 that "The predictive accuracy of the seven anti-PD-1 predictive transcriptome signatures did not improve, however, when these lesion-specific responses were included in the ROC analyses (data not shown)." We think it is critical for the authors to present the results with lesion-specific response information.

9. We fail to understand the rationale of the study design. The authors identified differential expression signature between HLA-A high vs low tumors among progressed patients (while controlling for cytolytic score), and find EMT markers and TNF-beta are associated with HLA-A expression. Why is it important to identify these genes to explain resistance to anti-PD1 while (1)

all these patient irrespective of HLA-A low or high develop resistance to (or progressed after) anti-PD1 treatment and (2) HLA-A expression is not associated with resistance in the dataset (only 5 out of 28 tumors in Figure 4A)?

Minor comments:

1. The authors tested several predictive signatures that failed to accurately define responding vs non-responding patients. However, one of the most predictive biomarker, the tumor mutational load, was not included in their analysis. The authors should discuss this limitation.
2. In Figure 3D, S1A, S1B, S2B the authors show a Pearson's correlation matrix with correlation coefficients, however, they do not report associated p-values to demonstrate whether these correlations are significant. We suggest to report Spearman rank correlation coefficients (that is more appropriate for non-parametric comparison), and (2) provide a supplemental table with associated p-values or encode it in their image (eg. varying sizes of dots).
3. In Figure 1A the authors describe the number of tumor biopsies or samples included in the study. The textbox refers to 'Non-responding patients' and below 'n=22'. We assume n=22 refers to the number of samples (as described in the figure legend) derived from non-responding patients. The authors should clearly indicate this in the Figure the number of patients and/or the number of samples.
4. Related to major comment 2: 'We did not identify any expressed alterations in the B2M, HLA-A, B, C in the transcriptome...' Could the authors describe how they identified expressed alterations and describe it in the methods section?
5. Johnson et al. (2016, Nature Communications) showed in tumor samples of melanoma patients that MHC-II positive tumors was elevated in responders, and showed better overall survival. Can the authors confirm/not confirm a similar trend in their dataset.
6. In the results section the authors write '91 melanoma tumors derived from 65 patients'. Figure 2B only shows 55 patients. We think the authors should include in Figure 2B the entire cohort of patients or justify exclusion of a subset of patients..
7. Could the authors add to the heatmap in Figure 3C the annotation as seen in Figure 2C, specifically it would be useful to see which patients in HLA(low) vs HLA(high) were responders vs non-responders?
8. The accession number should be available for review. In addition, the raw sequencing data should be made available through platforms like EGA (European Genome Archive). EGA provides controlled access to protect patient privacy, however, upon reasonable request the data can be accessed by other researchers. The availability of raw sequencing data could help other researchers in the field to build better predictors of patient responders vs non-responders.

Reviewer #2 (Remarks to the Author):

Authors have performed transcriptome analysis on longitudinal tumor biopsy material obtained from melanoma patients (n=91) who were treated with anti-PD1 antibody. Authors report that MHC class I downmodulation could be linked to anti-PD1 resistance and, this is associated with MITFlow/AXLhigh de-differentiated phenotype and cancer-associated fibroblast signatures. Authors further state that combination of anti-PD1 with drugs targeting TGFb pathway and/or the ones that reverses melanoma de-differentiation will be an effective therapeutic strategy.

Comments:

Authors have an impressive data on transcriptome analysis involving multiple tumor biopsies of patient undergoing anti-PD1 therapy. Unfortunately, supporting protein expression (FACS) data for MHC class I expression downmodulation are not entirely convincing. Tumor histology staining data on MHC-class I downmodulation and T-cell perforin/Granzyme staining will be ideal to convince the reader that HLA-class I is indeed down modulated at the tumor tissue level and lack of T-cell activity in such areas are thereof the cause of anti-PD1 therapy resistance.

Use of short term cell lines obtained after anti-PD1 to demonstrate de-differentiation markers are of some concerns due to change of cell phenotype in tissue cultures. Magnetic sorting of tumor cells directly from the biopsies will provide a unbiased picture.

Minor points:

1. Authors need to expand the some of the legends with greater details for easy understanding of the Figures. Provide legend keys for FACS data.
2. Band intensities in Western blot Figures need to be quantitated as bar graphs for easy comparison.

Reviewer #3 (Remarks to the Author):

I have read with great interest the study entitled 'Transcriptional downregulation of MHC class I and mesenchymal transition in melanoma resistance to PD-1 inhibition'. The authors have analyzed 91 biopsies from patients with melanoma that received PD-1 inhibitors using RNAsequencing, flow cytometry and in some cases functional studies. Although an impressively annotated cohort, many similar studies exist and are referenced. A more thorough bench-marking effort is warranted and more explanations needs to be provided in favor of some of the claims.

1. The hypothesis raised - that tumor heterogeneity is the reason for poor predictive accuracy of seven previously published transcriptome signature – is a reasonable explanation. This reviewer shares the view that currently we do not have a binary test to predict immunotherapy responses. This could be a main point of the manuscript, in which case it should be solidified, explained in relation to other studies, and mentioned in the title.

a. The claim can be solidified by performing ROC measuring the performance of each indicated signature in predicting PD-1 inhibitor responses in all respective patient cohorts where RNAseq data is publicly available. Ei, the TIDE, IMPRESS, 18-immune gene set etc signatures, how do they perform on the dataset of TIDE, IMPRESS, 18-immune gene set? If the current work is to impact the field such a benchmark is crucial.

b. Is there any possibility that differences in biopsy method can explain the heterogeneity of response in this dataset? TIDE, IMPRESS, 18-immune gene set etc are perhaps made on surgically resected biopsies and response evaluations were based on other target lesions. This dataset is also made on longitudinal biopsies of the same target lesions. Is there a possibility that a core biopsy can alter the fate of response to PD-1 inhibitors by altering the TME and inducing wound healing?

2. If the lack of predictive accuracy of known signature is not a major point then the data supporting the title should be scrutinized. The data is not entirely novel although the collected analyses in Figures 3-5 have some value in the present context.

a. This reviewer would prefer de-differentiated phenotype rather than mesenchymal transition. Melanoma is not an epithelial cancer so mesenchymal transition, although used frequently, is not so relevant.

b. In Table S3, Cell B2 and C2, have the labels been mixed up? If not, it is not easy to understand why HLA-A expression is lower in the HLA-A High column.

c. To complement Figure 4, it would be of value to observe the difference in HLA-1 expression between PRE and PROG/RES in those patients that have biopsies for both. The dots should be connected with lines so paired changes are visible.

- d. In relation to Fig 5d and discussion about tumor heterogeneity; more cell lines with high HLA-1 should be treated with vehicle or TGFb and then allowed to meet their autologous TILs. According to S5 the authors possess more cell lines. Will they all resist the TILs resulting in less death and IFNgamma if pre-treated with TGFbeta?
- e. In relation to Fig 5C, will knockdown or CRISPR deletion of SNAIL result in a diminished HLA-A downregulation or what is the signaling resulting in HLA-A downregulation?
- f. In Fig S5 there is an indication that IFNgamma suppresses SOX10, MLANA and MITF whilst inducing AXL in some cell lines. In these cell lines, are HLA-A levels down-regulated? If so, it conflicts with general dogma. If not, it conflicts with the general message of the study.

Response to Reviewers

REVIEWER 1

1. *The research question and hypothesis of the study is not clear. Is it MHC-I down-regulation or de-differentiation marker that is responsible for anti-PD1 resistance? In the first sentence of Discussion, the authors state “This study confirms that MHC class I downregulation associated with the de-differentiation phenotype is a hallmark of both innate and acquired resistance to PD-1 inhibitors.” We think the authors totally failed to provide sufficient support for this statement. If the de-differentiation phenotype or downregulation of MHC-I is the hallmark of both innate and acquired resistance to PD-1, please demonstrate that de-differentiation markers are sufficient to distinguish, at least, responders vs “nonresponders with innate resistance” among PRE-samples, or something comparable to that*

We would like to clarify several significant conclusions made in our manuscript that relate to this comment.

First, a key conclusion of our manuscript is that the accurate prediction of PD-1 inhibitor response based on a single pre-treatment biopsy is confounded by heterogeneity in tumor responses – we noted 16/68 patients had heterogenous tumour responses to PD1 inhibition (Page 13, paragraph 2 and Tables S1).

Second, in order to identify HLA-A downregulation as a resistance effector, we accounted for the heterogeneity in immune cell infiltration (i.e using the CYT score). In other words, *HLA-A* transcript downregulation was evident when analysed relative to CYT score (Page 15, second last paragraph) and resistance mechanisms reflect the degree of immune cell infiltration

Third, HLA-A downregulation was detected in 35-40% of PD1 pre-treatment and progression metastases (Page 16, last paragraph) – a common, but not ubiquitous mechanism of PD1 blockade resistance.

Considering these three key findings, we would not expect HLA-A downregulation to distinguish responders vs non-responders in the PRE-treatment samples. Indeed, none of the published predictive signatures accurately defined responders from PRE-treatment tumour analyses (Figure 1C), and we did not detect any significant differentially expressed gene signatures in the 44 pre-treatment tumors derived from responding and non-responding patients (page 13, first paragraph).

Taken together, we do not expect differentiation markers to distinguish between responders and non-responders.

2. *In continuation of comment 1, we fail to understand this statement “Not surprisingly, cell surface expression of HLA-A in the 12 PRE tumors did not reflect patient response, although 2 of 4 non-responding PRE tumors showed HLA-A downregulation (Patient 45 and 48; Figure 4A)” Why the authors study the down-regulation of MHC-I as mechanism for resistance to anti-PD1 while their low expression is not expected in non-responders? If down-regulation of HLA is such a determining factor, this should be reflected in the dataset that the authors analyze. Please demonstrate that the down-regulation of HLA is associated with better response in your dataset.*

Please see comments above Reviewer 1, point 1. We have updated this sentence to improve clarity and also tempered the conclusion as the small number of tumours analysed by flow cytometry. The sentence now reads:

We also examined the cell surface expression of HLA-A and HLA-DR, which was recently shown to correlate with response to PD1 inhibition (41), in the 15 PRE tumors. Although the tumor numbers are small and cell surface expression of HLA-ABC or HLA-DR did not accurately reflect patient response (Figure S7C & S7D).

3. *The authors compare several predictive scores/signatures of response to checkpoint inhibitors, but it is not clear how the AUROC was calculated. In particular, (i) how the positive and negative sets were determined, (ii) how the authors*

resolve the cases where one patient has multiple samples, (iii) how different signature values were calculated. We suggest the following to resolve the issue. (1) The methods to calculate the AUROC should be clearly stated. (2) We propose to present all signature values and positive/negative labels for each sample in your dataset as Supplementary Table.

We have added the following additional details on the ROC curve analysis in the Material and Methods section – Gene set and cell type enrichment analysis (Page 6, first paragraph).

These predictive scores were derived for each pre-treatment melanoma biopsy (n=44) and used with patient response data (complete (CR) and partial response (PR) versus stable (SD) and progressive disease (PD)) to generate receiver operator characteristic (ROC) curves in order to measure the performance of each indicated signature in predicting PD-1 inhibitor responses in our patient cohort. The performance of these seven PD-1 predictive signatures in predicting responding (irRC: CR and PR) were also evaluated in three publicly-available pre-treatment melanoma RNA-seq datasets with response data: (1) 49 patients treated with the PD-1 inhibitor, nivolumab (11), 26 patients treated PD-1 inhibitors nivolumab or pembrolizumab (17) and 41 patients treated with anti-CTLA4 (34). The area under the ROC curve was calculated using GraphPad version 8.2.1 using non-parametric estimates and 95% confidence interval based on the hybrid method of Wilson and Brown (35).

As indicated above, the ROC analysis was based on pre-treatment biopsies only – 44 patients each with a single PRE tumor. The ROC groups were responding (CR, PR) versus non-responding (PD, SD) patients. The data values for all seven predictive signatures are now included in Table S2

4. In addition, the authors claim that “For example, the innate PD-1 inhibitor resistance (IPRES) signature, which includes 26 genes associated with mesenchymal transition and BRAF/MEK inhibitor resistance, was associated with lack of PD-1 inhibitor response in pre-treatment melanoma biopsies in one study, but was not associated with PD-1 inhibitor response in other melanoma cohorts.” We agree that the validation of IPRES signature is limited, but we know other signatures such as IMPRES was demonstrated to be predictive in about ten independent melanoma datasets. Thus, we suggest to calculate AUROC with randomly picking up only one sample per patient in a given iteration, repeat this process for all possible configurations, and present the distribution of AUROC (mean and standard deviation).

We investigated the predictive value of seven published signatures in **pre-treatment** tumours, i.e one biopsy per patient (see also Reviewer 1, comment 3). There would be limited value in testing these predictive signatures in the PROG tumours, i.e tumors resected after progressing on PD1 inhibitor therapy.

5. The result that the authors present may be confounded with driver mutations. Bradley et al. (2015, Cancer Immunology Research) showed in melanoma cancer cell line model that BRAF-V600E mutations promote MHC-I internalization and re-location to the endolysosomal compartments. In Figure 4A, 4B cell surface expression are shown for patients with available flow cytometry data. Interestingly, patients 64 with the lowest HLA-ABC cell surface expression, but also patient 46 had a driver mutation in BRAF-V600E. Other patients with downregulated surface HLA-ABC had other mutations in MAPK pathway (48-NRAS Q61R, 49-NRAS Q61R, 45 -BRAF-V600K, 49-NRAS Q61R. This observation points to the possibility of other mechanisms (eg. non TGF-beta but mutation driven) of MHC-I surface downregulation. Could the authors include as a control, patients with low HLA-ABC surface expression and no mutations in genes involved in the MAPK pathway

The reviewer highlighted an important publication by Bradley et al. (Cancer Immunol Res) that showed BRAF^{V600E} can promote the internalisation of HLA-ABC from the cell surface. We have been able to add three additional BRAF/RAS wild type tumours to help address this query. We noted no significant differences in cell surface HLA-ABC expression according to melanoma genotype. These data have been included in Figure S7A and on page 16, last paragraph.

Eight of these eleven tumors had an activating BRAF or NRAS mutation, and although BRAF^{V600E} has been associated with the internalisation of HLA-ABC from the cell surface (41), we did not detect any genotype-associated differences in the cell surface expression of HLA-ABC in 31 melanoma tumors (Figure S7A).

We have also updated Figure 4B to include BRAF/NRAS wild type tumours with low HLA-ABC expression, and the tumor genotypes are now shown.

6. The authors write in the introduction section 'we demonstrate that non-genomic down-regulation of HLA class I expression, rather than complete loss of HLA class I molecules..' and discussion section 'We have now shown that non-genetic MHC class I downregulation occurred in..' The methods section only describes variant calling for single nucleotide variants (SNVs) and does not include insertions and deletions that could lead to potential frameshift mutations (eg. Zaretsky et al, 2016). Therefore, it is speculative to assume all HLA class I downregulation mechanisms are non-genomic or non-genetic. The authors should perform indel calling on their datasets and also discuss this limitation in the manuscript.

The reviewer highlights an important limitation. We only examined single nucleotide variants from the transcriptome sequence data because of the low reliability of indel alignment from RNA sequence data – this has been well reported (Engstrom et al. 2013: Nat Methods 10:1185 and Shukla et al. 2015: Nat Biotechnol 33:1152).

We have now removed the description of non-genomic HLA-downregulation and included details on the exclusion of indel calling in the methods 'Single nucleotide variant (SNV) analysis'.

Insertion and deletion calls were not included due to positional ambiguity and low alignment accuracy (35).

7. The authors explain in the methods section the estimation of the relative proportion of 22 types of infiltrating immune cells using CIBERSORT. In Figure S2C only CD8+ CIBERSORT extracted cells were compared between HLA-A(low) vs HLA-A(high). Could the authors perform a systematic analysis of the 22 types of immune cells and report the findings in a supplementary figure?

The CIBERSORT fractions of the 11 major leukocyte subsets have now been added in Figure S4B. We confirm that there was no significant difference between the fraction of immune cell subsets (as defined by CIBERSORT, see new Table S6) in the CYT score-matched tumors with low versus high HLA-A

The downregulation of HLA-A in these CYT-score matched tumors was not associated with diminished CD8+ T-cell content (Figure S4A), or alterations in the frequency of other leukocyte subsets (based on CIBERSORT profiling; Figure S4B and Table S6).

8. Many of the important data are not transparently presented. (1) Please show a scatter plot of CYT score and HLA-A expression in all tumors with different color-coding for PRE, RES, PROG tumors. (2) "It is worth noting that 16 melanoma PRE and PROG biopsies had matching flow cytometry and RNA sequence data and the HLA-ABC cell surface expression and HLA-A transcript expression were concordant in these samples (Spearman correlation 0.66, $p < 0.01$; data not shown)." We think this data should be presented. (3) The authors state on page 12 that "The predictive accuracy of the seven anti-PD-1 predictive transcriptome signatures did not improve, however, when these lesion-specific responses were included in the ROC analyses (data not shown)." We think it is critical for the authors to present the results with lesion-specific response information.

We have included the data requested by the reviewer.

Figure S3A now shows the CYT score vs HLA-A transcript scatter plot for all 79 tumors

Figure S7B now shows the concordance between HLA-A transcript expression and HLA-ABC cell surface expression in 16 melanoma biopsies; Spearman correlation 0.67, $p < 0.01$.

Figure S2A now shows the ROC curves for the seven immune response predictive signatures, with lesion specific responses included.

9. We fail to understand the rationale of the study design. The authors identified differential expression signature between HLA-A high vs low tumors among progressed patients (while controlling for cytolytic score), and find EMT markers and TNF-beta are associated with HLA-A expression. Why is it important to identify these genes to explain resistance to anti-PD1 while (1) all these patient irrespective of HLA-A low or high develop resistance to (or progressed after) anti-PD1 treatment and (2) HLA-A expression is not associated with resistance in the dataset (only 5 out of 28 tumors in Figure 4A)?

We are a little unclear about this comment.

It is true that HLA-A downregulation will not be the mechanism of resistance in all PD-1-resistant melanoma tumours – but heterogeneity of treatment resistance effectors is the norm in cancer (see review on melanoma BRAF/MEK inhibitor resistance: Johnson et al. Eur J Cancer 2015, 51:2792-2799)

It is also important to note that we know very little about anti PD-1-resistance effectors – only a few mechanisms have been defined and these are not common (i.e JAK1/2, B2M mutations; Zaretsky et al. NEJM 2016, 3759:819-829). The 30% of tumours with HLA-A downregulation is therefore an important addition to the field.

Minor comments

1. *The authors tested several predictive signatures that failed to accurately define responding vs non-responding patients. However, one of the most predictive biomarker, the tumor mutational load, was not included in their analysis. The authors should discuss this limitation.*

We have included additional discussion on the potential value of TMB and the limitation that we could not accurately estimate TMB without germline variant data (page 20, first paragraph).

It will be interesting to explore whether these limitations influence the predictive value of tumor mutation load, which has been shown to predict clinical benefit of immune checkpoint blockade in multiple cancers, including melanoma (71-73). Recent data suggest that tumor mutation burden may not change significantly during treatment (74), although intra-patient mutation burden heterogeneity has been observed (75). Further, when tumor samples were stratified according to melanoma subtype (i.e. cutaneous, occult, acral or mucosal), tumor mutation burden did not predict response to anti-PD-1 based immune checkpoint therapies (76). Unfortunately, germline variant data was not available in this study and it was not possible to accurately estimate tumor mutation burden in this study,

2. *In Figure 3D, S1A, S1B, S2B the authors show a Pearson's correlation matrix with correlation coefficients, however, they do not report associated p-values to demonstrate whether these correlations are significant. We suggest to report Spearman rank correlation coefficients (that is more appropriate for non-parametric comparison), and (2) provide a supplemental table with associated p-values or encode it in their image (eg. varying sizes of dots).*

We have updated the figures to show the Spearman rank correlation as suggested. The FDR-adjusted p-value was <0.01 for all signatures shown in these figures, and this information has now been included in each Figure legend and the new Table S7.

3. *In Figure 1A the authors describe the number of tumor biopsies or samples included in the study. The textbox refers to 'Non-responding patients' and below 'n=22'. We assume n=22 refers to the number of samples (as described in the figure legend) derived from non-responding patients. The authors should clearly indicate this in the Figure the number of patients and/or the number of samples.*

Figure 1A has been updated and both number of patients and number of samples are indicated.

4. *Related to major comment 2: 'We did not identify any expressed alterations in the B2M, HLA-A, B, C in the transcriptome...' Could the authors describe how they identified expressed alterations and describe it in the methods section?*

We have expanded the description of variant calling in the Material and Methods: Single nucleotide variant (SNV) analysis:

SNVs were called against the reference genome using VarScan2. Minimum variant frequency was set to 20% and other parameters were left at their default values. Briefly, the SAMtools mpileup utility provided a summary of the read coverage, and the mpileup output was processed using VarScan 2 to call variants and produce a VCF format file with variants that passed the minimum read and allele frequency thresholds. Insertion and deletion calls were not included due to positional ambiguity and low alignment accuracy (35). Visualisation of the resulting VCF files and analysis was performed through the use of Ingenuity Variant Analysis software (<https://www.qiagenbioinformatics.com/products/ingenuity-variant-analysis>) from Qiagen, Inc.

5. Johnson et al. (2016, Nature Communications) showed in tumor samples of melanoma patients that MHC-II positive tumors was elevated in responders, and showed better overall survival. Can the authors confirm/not confirm a similar trend in their dataset.

We have included the MHC II expression data for our sample set and expression at pre-treatment does not predict response. These data are now shown in Figure S7D and described in page 17, first paragraph.

We also examined the cell surface expression of HLA-ABC and HLA-DR, which was recently shown to correlate with response to PD-1 inhibition (42), in the 15 PRE tumors. Although the tumor numbers were small, the cell surface expression of HLA-ABC or HLA-DR at PRE did not accurately reflect patient response (Figure S7C & S7D).

6. In the results section the authors write '91 melanoma tumors derived from 65 patients'. Figure 2B only shows 55 patients. We think the authors should include in Figure 2B the entire cohort of patients or justify exclusion of a subset of patients..

Figure 2B displays the relationship between longitudinal tumor samples (biopsy site), patient response and the transcriptome cytolytic score (CYT score). RNA sequence data was only available for these 79 tumor samples. This was mentioned in the Results: Patient and tumor characteristics section (page 10, paragraph 1), has been updated to improve clarity. We have also updated Table S1 to clearly indicate samples with flow cytometry and RNA seq data.

Transcriptome analysis was performed on RNA sequence data (n = 79 tumors; 55 patients) and flow cytometric analysis on single cell suspensions (n = 31; 24 patients) from a total of 94 melanoma tumors derived from 68 patients treated with PD-1 inhibitor monotherapy.

7. Could the authors add to the heatmap in Figure 3C the annotation as seen in Figure 2C, specifically it would be useful to see which patients in HLA(low) vs HLA(high) were responders vs non-responders?

Figure 3C has been updated to show the best irRC data for each patient.

8. The accession number should be available for review. In addition, the raw sequencing data should be made available through platforms like EGA (European Genome Archive). EGA provides controlled access to protect patient privacy, however, upon reasonable request the data can be accessed by other researchers. The availability of raw sequencing data could help other researchers in the field to build better predictors of patient responders vs non-responders

The RNA sequence data has been deposited in the European Genome-phenome Archive under accession number EGAS00001001552 and dataset accession EGAD00001005738 – these details are provided in the manuscript in the 'Data availability' section.

REVIEWER 2

1. Authors have an impressive data on transcriptome analysis involving multiple tumor biopsies of patient undergoing anti-PD1 therapy. Unfortunately, supporting protein expression (FACS) data for MHC class I expression downmodulation are not entirely convincing. Tumor histology staining data on MHC-class I downmodulation and T-cell perforin/Granzyme staining will be ideal to convince the reader that HLA-class I is indeed down modulated at the tumor tissue level and lack of T-cell activity in such areas are thereof the cause of anti-PD1 therapy resistance.

We understand that tumor immunohistochemistry is commonly used as a means of exploring protein accumulation, and does have the advantage of preserving architecture and spatial distribution. However, immunohistochemistry results have previously been reported as significantly discordant when blindly scored by two experienced pathologists (Johnson 2016 – ref 42), attesting to the largely subjective nature of the assay. In contrast, our analysis allowed for the precise quantification of HLA-ABC expression in melanoma cells relative to infiltrating lymphocytes from the same biopsy. This could only be achieved with FACS – which is sensitive and quantitative over a broad dynamic range, and robust as over 10,000 cells per FACS experiment were analysed. Further, our FACS data for HLA-ABC expression analysed by flow

cytometry was significantly positively correlated and validated by the transcriptome data – and these correlation data have now been included in new Figure S7B, see Reviewer 1, major point 8.

2. *Use of short term cell lines obtained after anti-PD1 to demonstrate de-differentiation markers are of some concerns due to change of cell phenotype in tissue cultures. Magnetic sorting of tumor cells directly from the biopsies will provide a unbiased picture.*

The reviewer is concerned that cell culturing has influenced the dedifferentiation phenotype in the melanoma cell models used in this report. We have reviewed the consistency of the dedifferentiation phenotype between the original tumor and the derived cell line in 12 tumor-cell line pairs. We found that the AXL^{High}/MITF^{Low} melanoma cell lines were commonly derived from NGFR^{High} tumours, whereas AXL^{Low}/MITF^{High} melanoma cells were derived from NGFR^{Low} melanoma tumors (NGFR is a dedifferentiation marker used in flow cytometry of tumor samples, that is positively correlated with AXL, and negatively correlated with MITF and the melanocytic antigens MART-1 and gp100; Hoek et al. 2008 Cancer Research; Muller et al. Nature Communications; Tirosh et al 2016 Science). We have added a sentence to indicate the phenotype concordance (Page 17, third paragraph sentence shown below) and included the data in the new Figure S7B.

Importantly, we confirmed that the melanoma de-differentiation phenotype was usually concordant between the tumor biopsy (defined by NGFR positivity) and the corresponding PD1 PROG cells models (Figure S8B).

We also wish to highlight that our conclusions confirming that melanoma dedifferentiation is associated with diminished HLA-class I expression is not only based on cell models, but derived from transcriptome analysis of fresh tumour dissociates.

Minor comments

1. *Minor point: Authors need to expand the some of the legends with greater details for easy understanding of the Figures. Provide legend keys for FACS data*

We have reviewed the figure legends and added additional details where required, and included FACS legend details in all figures.

2. *Minor point: Band intensities in Western blot Figures need to be quantitated as bar graphs for easy comparison*

We have included the quantitation of band intensities from Figure 5C and Figure S8 in the new Figures S9 and S10.

REVIEWER 3

1a. *The hypothesis raised - that tumor heterogeneity is the reason for poor predictive accuracy of seven previously published transcriptome signature – is a reasonable explanation. This reviewer shares the view that currently we do not have a binary test to predict immunotherapy responses. This could be a main point of the manuscript, in which case it should be solidified, explained in relation to other studies, and mentioned in the title. a. The claim can be solidified by performing ROC measuring the performance of each indicated signature in predicting PD-1 inhibitor responses in all respective patient cohorts where RNAseq data is publicly available. Ei, the TIDE, IMPRESS, 18-immune gene set etc signatures, how do they perform on the dataset of TIDE, IMPRESS, 18-immune gene set? If the current work is to impact the field such a benchmark is crucial.*

We have added analysed the predictive performance of the seven predictive transcriptome signatures on three additional melanoma RNA-seq data sets (from Van Allen, Hugo and Riaz). These details have been added to the 'Materials and Methods: Gene set and cell type enrichment analysis': page 6

The performance of these seven PD-1 predictive signatures in predicting responding (irRC: CR and PR) were also evaluated in three publicly-available pre-treatment melanoma RNA-seq datasets with response data: (1) 49 patients treated with the PD-1 inhibitor, nivolumab (11), 26 patients treated PD-1 inhibitors nivolumab or pembrolizumab (17) and 41 patients treated with anti-CTLA4 (34).

The ROC curve analysis for these datasets is presented in Figure S1, including the AUC and P-values. As shown in Figure S1 and described in the Results section, page 13, paragraph 1:

Additionally, we found that none of these predictive transcriptomic signatures were consistently and significantly associated with irRC response in three separate immune-checkpoint inhibitor treated melanoma patient RNA-seq datasets (Figure S1).

1b. *Is there any possibility that differences in biopsy method can explain the heterogeneity of response in this dataset? TIDE, IMPRESS, 18-immune gene set etc are perhaps made on surgically resected biopsies and response evaluations were based on other target lesions. This dataset is also made on longitudinal biopsies of the same target lesions. Is there a possibility that a core biopsy can alter the fate of response to PD-1 inhibitors by altering the TME and inducing wound healing?*

It is important to clarify a few key points regarding this query.

First, the response signatures [TIDE, IMPRES, 18-immune gene set etc] were tested in pre-treatment tumors – and the longitudinal, on-treatment tumors were not part of the predict section of our manuscript. As stated in first paragraph, page 13:

We initially examined the predictive accuracy of seven transcriptome signatures associated with clinical response to PD-1 inhibition (11,15,17-19,31,32) in the 44 pre-treatment tumors with available RNA sequence data.

Second, the query regarding core biopsies influencing heterogeneity of response is difficult to conclusively address, although it is worth noting that core biopsies were uncommon in our cohort (5/44 pre-treatment tumours were cored), and that the lesion response of tumors that had been core-biopsied were variable with 3 responding lesions (CR, PR, PR) and 2 non-responding lesion (PD, SD) (see Table 2). Finally, we have updated the discussion to provide additional details on intra-patient heterogeneity of response to immune checkpoint inhibition (page 19, last paragraph).

In addition to intra-patient heterogeneity in tumor inflammation, we also observed significant variation in the response of individual lesions to immune checkpoint inhibition within melanoma patients, with heterogeneity of tumour responses observed in 85% and 69% of melanoma patients who did not achieve an objective response to pembrolizumab monotherapy or combination ipilimumab plus nivolumab, respectively (38; 66).

2. If the lack of predictive accuracy of known signature is not a major point then the data supporting the title should be scrutinized. The data is not entirely novel although the collected analyses in Figures 3-5 have some value in the present context

We believe that the lack of predictive accuracy of existing signatures is an important contribution to the field and have updated the title of our manuscript to broadly reflect the content:

Heterogeneity of melanoma responses limits the accuracy of PD-1 inhibitor predictive signatures

2a *This reviewer would prefer de-differentiated phenotype rather than mesenchymal transition. Melanoma is not an epithelial cancer so mesenchymal transition, although used frequently, is not so relevant.*

We have adopted de-differentiated throughout the manuscript.

2b. *In Table S3, Cell B2 and C2, have the labels been mixed up? If not, it is not easy to understand why HLA-A expression is lower in the HLA-A High column.*

The labels in cells B2 and C2 have been corrected.

- 2c. *To complement Figure 4, it would be of value to observe the difference in HLA-1 expression between PRE and PROG/RES in those patients that have biopsies for both. The dots should be connected with lines so paired changes are visible.*

We have added dotted lines to connect patient-matched PRE and PROG tumors in Figure 4 and updated the Figure legend accordingly.

- 2d. *In relation to Fig 5d and discussion about tumor heterogeneity; more cell lines with high HLA-1 should be treated with vehicle or TGF β and then allowed to meet their autologous TILs. According to S5 the authors possess more cell lines. Will they all resist the TILs resulting in less death and IFN γ if pre-treated with TGF β ?*

We have included three melanoma cell lines to show that TGF β downregulates HLA-A expression, and that this leads to reduced T cell activation in immune cell: autologous melanoma cell cultures. These data are shown in the updated Figure 5B-5D.

Although we have generated many short-term melanoma cell models, not all have autologous immune cells, and few show immune cell activation in melanoma-immune cell co-cultures. We utilised the more sensitive flow cytometry-based assay to detect IFN γ production for the updated Figure 5D, and the method has been added to the 'co-culture melanoma:TIL assay', page 7 of the Materials and Methods.

- 2e *In relation to Fig 5C, will knockdown or CRISPR deletion of SNAIL result in a diminished HLA-A downregulation or what is the signaling resulting in HLA-A downregulation?*

We attempted to downregulate SNAIL with lentiviral transduced shRNA. Although the transduction efficiencies were >80%, we were not able to achieve effective SNAIL silencing in the presence of TGF β exposure.

- 2f *In Fig S5 there is an indication that IFN γ suppresses SOX10, MLANA and MITF whilst inducing AXL in some cell lines. In these cell lines, are HLA-A levels down-regulated? If so, it conflicts with general dogma. If not, it conflicts with the general message of the study.*

Interferon gamma does not consistently alter the levels of SOX10, MITF or MLANA in our cell models. It is apparent that AXL is reduced in a few cell lines in response to interferon gamma, but this is not a consistent finding, and thus not highlighted in our report. Collectively, interferon gamma does not promote the de-differentiated phenotype we observe in some PROG cell models. Quantitation for these Westerns is now included in the new Figure S9.

REVIEWERS' COMMENTS:

Reviewer #1 (Remarks to the Author):

Major comments:

We find the authors have improved the manuscript and sufficiently addressed our previous comments. However, changes in the title ('Heterogeneity of melanoma responses limits the accuracy of PD-1 inhibitor predictive signatures'), signal the author's intention to change the focus of the manuscript. We are extremely surprised to see such a change at this stage of the review process – I would say this is quite unprecedented. Indeed, we agree that studying the effects of intra-patient heterogeneity of lesion response and single biopsy of PRE-treatment samples and its implications on building robust predictors is an important question to ask, but to properly substantiate this important challenge we are missing key analyses experiments (see below) that would merit an independent publication and significantly more work. Therefore, we advise the authors to keep the original focus and direction of the manuscript. If the authors choose to go with the new focus, they need to:

(1) Test the predictive accuracy of immune signatures on the ON-treatment biopsies. Are there differences in the predictive power when different lesions are considered?

(2) In order to support the claim that heterogeneity of lesion response limits the accuracy of predictive signatures, the authors would need to have a collection of PRE-treatment biopsies sampled from different sites. First, it would be necessary to understand and characterize the heterogeneity among the different biopsy sites and second the authors would need test whether this has an effect on the predictive accuracy of the immune signatures.

(3) We don't understand the author's reasoning to include publicly available data sources (Hugo et al., Van Allen et al., Riaz et al.), that were not properly (experimentally) designed to support their claims (none of these data sources had PRE-treatment biopsies of different sites). The lack of predictive accuracy of immune signatures (if true) is of course concerning and worthwhile exploring, but it requires careful normalization and processing of data, which the authors negligibly fail to describe and provide (?!?). Even more so, as their analysis contradicts numerous published reports in the field. If the authors wish to include the comparison to the original publications, this very strong claim obviously needs to be comprehensively and rigorously tested and compared to the original findings in a way more exhaustive and careful manner. Specifically, we will need to see:

a. Can the authors exclude that variation in sequencing technology, differences in data pre-processing and normalization may affect the results of the predictive signatures tested? To answer this, the authors need to report how they obtained the data from the three publicly available data sources (only for Van Allen (TPM) was the normalization reported). How was the data normalized (RPKM, TPM, FPKM or Raw reads)? Differences in normalization can have significant differences in performance of immune signatures (in particular for methods that rely on preservation of relative ranking of genes within a sample). The authors should test the same normalization for the different datasets and additionally test different data normalization procedures across predictive signatures and verify if this alone changes the predictive accuracy. Furthermore, the authors should test different normalization procedures on their own dataset and see if this improves predictive accuracy.

b. We noticed differences in how response is defined compared with the original publications that obviously raise questions (Riaz, Hugo, Van Allen). For instance, in Figure S1 some SD patients are grouped as responders. Please verify that for each analysis a consistent definition of responding patients is defined.

Minor comment

In line 353 and 590 the authors use the expression "tumor heterogeneity". This is can be potentially very confusing for the reader. Please change it to "intra-patient heterogeneity".

Reviewer #2 (Remarks to the Author):

Manuscript is much improved after revisions.

It is unfortunate that authors have not adequately addressed the original request for tumor histology staining for HLA-class I downmodulation. It is hard to buy the explanation that FACS are more sensitive than histology techniques. CYTOF combined with digital imaging technologies can remove any subjective assessment of the tissue sections.

Reviewer #3 (Remarks to the Author):

The new data, especially the benchmarking of different predictive algorithms, are very important. As these data refutes many papers in high-impact journal I hope these data will stand the test of time. I want to thank the authors for doing this important analysis, I can really appreciate the effort.

/Jonas Nilsson

REVIEWER 1

We have carefully considered this reviewer's comments regarding the change of manuscript title. We can assure the editors that the focus of the work has not changed, although we understand the confusion.

The change in title was our response to some of the comments made initially by Reviewer 3. In particular:

1a The hypothesis raised - that tumor heterogeneity is the reason for poor predictive accuracy of seven previously published transcriptome signature – is a reasonable explanation. This reviewer shares the view that currently we do not have a binary test to predict immunotherapy responses. This could be a main point of the manuscript, in which case it should be solidified, explained in relation to other studies, and mentioned in the title. The claim can be solidified by performing ROC measuring the performance of each indicated signature in predicting PD-1 inhibitor responses in all respective patient cohorts where RNAseq data is publicly available.

In our resubmission we included the requested analyses of additional predictive transcriptome signatures (new Figure S1C), but, in hindsight, this did not necessitate a title change. In fact, the majority of our manuscript revisions (new and revised Figures 4B, 5B-5D, S3A, S7A-S7D) supported the role of MHC class I downregulation and de-differentiation in immunotherapy resistance and we should have updated, rather than changed the title.

Considering all these details, we believe that the following manuscript title is an appropriate and accurate reflection of our work, and we would like to update the manuscript accordingly.

Transcriptional downregulation of MHC class I and melanoma de-differentiation in resistance to PD-1 inhibition

With this title update and clarification on the focus of our work the additional (1) and (2) comments made by Reviewer 1 are no longer relevant and we will not address these any further.

We would, however, like to carefully consider several issues raised by this reviewer in point (3)

(3a) We don't understand the author's reasoning to include publicly available data sources (Hugo et al., Van Allen et al., Riaz et al.), that were not properly (experimentally) designed to support their claims (none of these data sources had PRE-treatment biopsies of different sites). The lack of predictive accuracy of immune signatures (if true) is of course concerning and worthwhile exploring, but it requires careful normalization and processing of data, which the authors negligibly fail to describe and provide (!?!). Even more so, as their analysis contradicts numerous published reports in the field. If the authors wish to include the comparison to the original publications, this very strong claim obviously needs to be comprehensively and rigorously tested and compared to the original findings in a way more exhaustive and careful manner. Specifically, we will need to see: a. Can the authors exclude that variation in sequencing technology, differences in data pre-processing and normalization may affect the results of the predictive signatures tested? To answer this, the authors need to report how

they obtained the data from the three publicly available data sources (only for Van Allen (TPM) was the normalization reported). How was the data normalized (RPKM, TPM, FPKM or Raw reads)? Differences in normalization can have significant differences in performance of immune signatures (in particular for methods that rely on preservation of relative ranking of genes within a sample). The authors should test the same normalization for the different datasets and additionally test different data normalization procedures across predictive signatures and verify if this alone changes the predictive accuracy. Furthermore, the authors should test different normalization procedures on their own dataset and see if this improves predictive accuracy.

First, our claim was that none of the 'tested predictive transcriptomic signatures were consistently and significantly associated with irRC response'. This is correct based on the analyses of our cohort and several publicly available data sets (Figure 1C and S1). We hypothesised that the poor predictive value of these signatures may be due to heterogeneity of response, and tested this assumption in our cohort, with no improvement in predictive accuracy (Figure S2A). The inclusion of publicly available data sources was to highlight the lack of accurate prediction with current signatures, rather than to examine the impact of heterogeneity of response. As the reviewer indicated, few data sources (if any) would include several PRE-treatment core specimens from multiple sites and with lesion response data.

Second, the comment that our work contradicts many published reports does not accurately reflect the latest information – with many reports unable to validate published signatures. We provide a few pertinent examples for the editors.

- a. Tumour mutation burden (when stratified by melanoma subtype) and individual immune cell subsets (based on CIBERSORT) did not predict melanoma patients responding to PD-1 based therapy in a large melanoma cohort of 206 patients (Liu, 2019. Nat Med 25:1916)
- b. The IPRES signature was not associated with PD-1 inhibitor response in other melanoma cohorts (Riaz, 2017 Cell 171:934; Chen, 2016 Cancer Discov 6:827).
- c. The validity of IMPRES to predict immune checkpoint blockade in melanoma is contentious (Carter, 2019 Nature Med 25, 1833-1835)
- d. A combined comparison of the predictive accuracy of several transcriptome-based predictors in PD-1 pre-treatment tumors confirmed that few signatures consistently predicted response to immunotherapy (Table S6; Auslander, 2018 Nat Med 24:1545)

Third, the question of sequence normalisation is an important one, and we have compared both FPKM (which accounts for gene length) and TMM-Voom normalised transcriptome data in our cohort (n=44, PRE-treatment biopsies). Regardless of the normalisation approach, the ROC AUC data are similar with none of the signatures accurately predicting response (CR/PR vs PD/SD). The comparison AUC and p values data from ROC analysis are tabulated below.

Signature	TMM-Voom		FPKM	
	AUC	p	AUC	p
IPRES ₁			0.5372	0.6727
IMPRES	0.5155	0.8603	0.5537	0.5417
CYT score	0.5475	0.5893	0.5083	0.9252
TIDE	0.6694	0.0543	0.6591	0.0707
18-immune gene set ₁			0.5579	0.5110
CD8A/CSF1R ratio	0.6570	0.0744	0.6095	0.2135
CD8+ T cells CIBERSORT ₂			0.6136	0.1967

¹FPKM used only as ssGSEA used for these signatures requires gene expression data that is normalisation for gene length

²FPKM used only as gene expression data required in non-log linear space

Fourth, regarding normalisation of previously published transcriptome datasets. We have outlined the source of these datasets (see Figure S1 legend, below). These datasets have been uploaded by the authors as normalised data. As such, we did not outline these normalisation details which are described in the relevant cited publication (but could

include additional details if necessary). We suggest we add the statement that the datasets were downloaded as normalised data from GSE78220, GSE91061 etc in the figure legend and in the methods section.

Figure S1 Performance of immune-predictive transcriptome scores

Immune-predictive transcriptome scores derived for three publicly available immune checkpoint inhibitor melanoma datasets (GSE78220, GSE91061 and TPM-RSEM values for the Van Allen dataset (34) from GitHub: https://github.com/vanallenlab/VanAllen_CTLA4_Science_RNASeq_TPM. PRE-treatment melanoma biopsies and patient response data (CR/PR vs SD/PD) were used to generate receiver operator characteristic (ROC) curves measuring the performance of each indicated signature in predicting PD-1 inhibitor responses in 49 patients treated with the PD-1 inhibitor, nivolumab (11), 26 patients treated PD-1 inhibitors nivolumab or pembrolizumab (17) and 41 patients treated with anti-CTLA4 with RECIST data (34)

- b. *We noticed differences in how response is defined compared with the original publications that obviously raise questions (Riaz, Hugo, Van Allen). For instance, in Figure S1 some SD patients are grouped as responders. Please verify that for each analysis a consistent definition of responding patients is defined.*

We have been very specific regarding our definition of responders vs non-responders (see Methods: 'Gene set and cell type enrichment analysis' - text below).

These predictive scores were derived for each pre-treatment melanoma biopsy (n=44) and used with patient response data (complete (CR) and partial response (PR) versus stable (SD) and progressive disease (PD)) to generate receiver operator characteristic (ROC) curves in order to measure the performance of each indicated signature in predicting PD-1 inhibitor responses in our patient cohort. The performance of these seven PD-1 predictive signatures in predicting responding (irRC: CR and PR) were also evaluated in three publicly-available pre-treatment melanoma RNA-seq datasets with response data:

The reviewer noticed that the Van Allen cohort in Figure S1 was analysed according to the above criteria but also according to clinical benefit (CR, PR or SD with overall survival > 1 month) vs no clinical benefit (PD or SD with overall survival < 1 year). This additional analysis was only done for the Van Allen dataset, because the original manuscript stratified the patients in this manner. These details were provided within the Figure.

Minor comments: In line 353 and 590 the authors use the expression "tumor heterogeneity". This is can be potentially very confusing for the reader. Please change it to "intra-patient heterogeneity".

We are happy to change the text as suggested

REVIEWER 2

1. *It is unfortunate that authors have not adequately addressed the original request for tumor histology staining for HLA-class I downmodulation. It is hard to buy the explanation that FACS are more sensitive than histology techniques. CYTOF combined with digital imaging technologies can remove any subjective assessment of the tissue sections.*

We appreciate that imaging mass cytometry would have been a valuable addition, unfortunately, we did not have the capacity to undertake this type of histology. We had assumed the reviewer meant standard tumor immunohistochemistry in his initial comments, and we felt this would not be a valuable addition.

In summary, we appreciated the thoughtful comments made by the reviewer's, and would like the opportunity to make the additional revisions described above.

Yours sincerely